# The Meta-Representation Hypothesis

## Abstract

Humans rely on high-level understandings of things, i.e., meta-representations, to engage in abstract reasoning. In complex cognitive tasks, these meta-representations help individuals abstract general rules from experience. However, constructing such meta-representations from high-dimensional observations remains a longstanding challenge for reinforcement learning (RL) agents. For instance, a well-trained agent often fails to generalize to even minor variations of the same task, such as changes in background color, while humans can easily handle. In this paper, we theoretically investigate how meta-representations contribute to the generalization ability of RL agents, demonstrating that learning meta-representations from high-dimensional observations enhance an agent's ability to generalize across varied environments. We further hypothesize that deep mutual learning (DML) among agents can help them learn the meta-representations that capture the underlying essence of the task. Empirical results provide strong support for both our theory and hypothesis. Overall, this work provides a new perspective on the generalization of deep reinforcement learning.

**Project website:** Click **here** to view our website.

## 1. Introduction

A meta-representation refers to a higher-order form of representation—essentially, a representation of a representation (Wilson, 2012; Redshaw, 2014). In other words, it is an abstraction that captures not just the content of an experience or concept, but how that content is represented. To illustrate, consider the saying, "*There are a thousand Hamlets in a thousand people's eyes.*" Here, the text of *Hamlet* serves as a direct representation, whereas each reader's interpretation

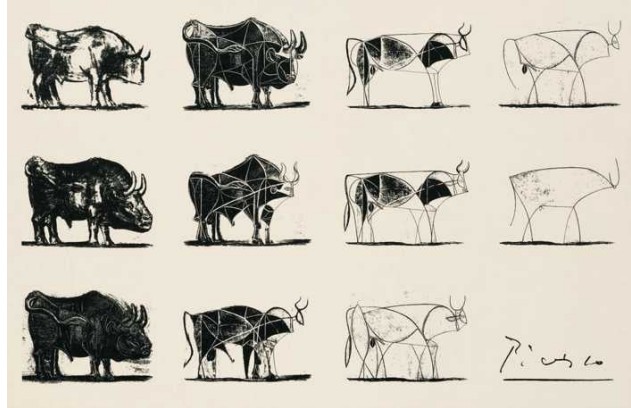

Figure 1. Pablo Picasso's *The Bull* (Scott, 2019). By focusing on and exaggerating specific details, rather than trying to capture every detail realistically, artists can convey the core meaning or essence of the subject—*meta-representation*, more powerfully.

of the play is a form of meta-representation—an abstract, high-level understanding of the content.

Humans process and integrate vast amounts of information from the real world through meta-representations, which are underlying structured information beyond the direct sensory representations of things we perceive (Figure 1). These meta-representations enable us to generalize across tasks with similar underlying semantics. For instance, once we have learned how to play a video game, we can apply the same skills even if the game's visual presentation changes. This suggests that the ability to perform tasks is not tied to the specific visual details of the game, but to the underlying cognitive processes that abstract away from these changes. The development of abstract thinking is linked to the human prefrontal cortex (Bengtsson et al., 2009; Dumontheil, 2014), and certain inhibitory neurons further enhance the brain's processing efficiency (Pi et al., 2013).

While humans can generalize across tasks by relying on abstract meta-representations, visual reinforcement learning (VRL) faces a significant challenge in this regard. Although well-trained agents can solve complex tasks, they often struggle to transfer their experience to new environments. Even subtle changes, such as variations in scene colors, can hinder their ability to generalize, demonstrating that

[1]Anonymous Institution, Anonymous City, Anonymous Region, Anonymous Country. Correspondence to: Anonymous Author <anon.email@domain.com>.

Preliminary work. Under review by the International Conference on Machine Learning (ICML). Do not distribute.

**Algorithm 1** MDP Generator
---
1: **Initialize:** Underlying MDP $\mathcal{M}$ and behavior policy $\pi$
2: **while** collecting data **do**
3:     Randomly initialize a rendering function $f$
4:     Underlying initial state $s_0 \sim \mathcal{M}$
5:     **for** $t = 0$ **to** $T$ **do**
6:         The noisy observation $o_t = f(s_t)$
7:         Choose action $a_t \sim \pi(\cdot|o_t)$
8:         Update environment $r_t, s_{t+1} \sim \mathcal{M}(s_t, a_t)$
9:         Store data $(o_t, a_t, r_t)$
10:     **end for**
11: **end while**

their learning is overly dependent on specific visual inputs (Cobbe et al., 2019; 2020).

What makes it difficult for reinforcement learning agents to generalize? How can these agents develop the ability to construct meta-representations just like humans?

Our central theory, presented in Algorithm 1, assumes the existence of several Markov Decision Processes (MDPs) sharing an underlying MDP. Imagine the scenario of building a reinforcement learning benchmark to test the generalization performance of algorithms. We would first implement the core code for the underlying MDP $\mathcal{M}$, which reflects the intrinsic properties of the task. Then we randomly initialize a rendering function $f$, which obfuscates the underlying state $s_t$ into the agent's observation $o_t = f(s_t)$, akin to how different schools of painters might depict the same scene in various styles. To achieve good generalization, the agent must learn to ignore the interference from $f$. In this scenario, learning the meta-representation means that the agent has learned to perceive beyond the noisy observation $o_t$ and grasp the true underlying state $s_t$. This process is far more challenging than simply achieving high performance during training, as it requires the agent to filter out the noise and focus on the core task structure.

This paper aims to develop a theory of generalization in reinforcement learning, with a particular focus on learning meta-representations that capture the essential structure of tasks beyond superficial observations. Unlike traditional approaches such as the Partially Observable Markov Decision Process (POMDP) (Murphy, 2000), which focuses primarily on the challenge of partial observability of the true state, our framework emphasizes the ability of agents to learn abstract, high-level representations beyond the noisy observations. Furthermore, our meta-representation hypothesis posits that deep mutual learning (DML) (Zhang et al., 2018b) between agents can facilitate the learning of these meta-representations, thereby improving generalization performance across different environments sharing the same underline semantics. Extensive experiments on Procgen

(Cobbe et al., 2019; 2020) support our theory and hypothesis, demonstrating that PPO (Schulman et al., 2017) with our DML-based framework achieves significant improvements over the standard PPO.

Overall, this study presents a novel perspective on improving the generalization capabilities of deep reinforcement learning, providing insights that could contribute to more adaptable and robust decision-making in diverse and dynamic environments. The main contributions of this paper are summarized as follows:

- We theoretically prove that improving the policy robustness to irrelevant features enhances generalization performance. To the best of our knowledge, we are the first to provide a rigorous proof of this intuition.

- We propose a hypothesis that deep mutual learning (DML) can facilitate the learning of meta-representations by agents, and we also provide intuitive insights to support this hypothesis.

- Strong empirical results support our theory and hypothesis, showing that DML technique leads to consistent improvements in generalization performance.

## 2. Preliminaries

In this section, we introduce reinforcement learning under the generalization setting in Section 2.1, as well as the DML technique in Section 2.2.

### 2.1. Markov Decision Process and Generalization

Markov Decision Process (MDP) is a mathematical framework for sequential decision-making, which is defined by a tuple $\mathcal{M} = (\mathcal{S}, \mathcal{A}, r, \mathcal{P}, \rho, \gamma)$, where $\mathcal{S}$ and $\mathcal{A}$ represent the state space and action space, $r : \mathcal{S} \times \mathcal{A} \mapsto \mathbb{R}$ is the reward function, $\mathcal{P} : \mathcal{S} \times \mathcal{A} \times \mathcal{S} \mapsto [0, 1]$ is the dynamics, $\rho : \mathcal{S} \mapsto [0, 1]$ is the initial state distribution, and $\gamma \in (0, 1)$ is the discount factor.

Define a policy $\mu : \mathcal{S} \times \mathcal{A} \mapsto [0, 1]$, the action-value function and value function are defined as

$$Q^\mu(s_t, a_t) = \mathbb{E}_\mu \left[ \sum_{k=0}^\infty \gamma^k r(s_{t+k}, a_{t+k}) \right],$$
$$V^\mu(s_t) = \mathbb{E}_{a_t \sim \mu(\cdot|s_t)} \left[ Q^\mu(s_t, a_t) \right]. \tag{1}$$

Given $Q^\mu$ and $V^\mu$, the advantage function can be expressed as $A^\mu(s_t, a_t) = Q^\mu(s_t, a_t) - V^\mu(s_t)$.

In our generalization setting, we introduce a rendering function $f : \mathcal{S} \mapsto \mathcal{O}_f \subset \mathcal{O}$ to obfuscate the agent's actual observations, which is a bijection from $\mathcal{S}$ to $\mathcal{O}_f$. We now define the MDP induced by the underlying MDP $\mathcal{M}$ and the rendering function $f$, denote it as $\mathcal{M}_f = (\mathcal{O}_f, \mathcal{A}, r_f, \mathcal{P}_f, \rho_f, \gamma)$,

where $\mathcal{O}_f$ represents the observation space, $r_f : \mathcal{O}_f \times \mathcal{A} \mapsto \mathbb{R}$ is the reward function, $\mathcal{P}_f : \mathcal{O}_f \times \mathcal{A} \times \mathcal{O}_f \mapsto [0, 1]$ is the dynamics, and $\rho_f : \mathcal{O}_f \mapsto [0, 1]$ is the initial observation distribution. We present the following assumptions:

**Assumption 2.1.** Assume that $f$ can be sampled from a distribution $p : \mathcal{F} \mapsto [0, 1]$, where $f \in \mathcal{F}$.

**Assumption 2.2.** Given any $f \in \mathcal{F}$, $o_0^f, o_t^f, o_{t+1}^f \in \mathcal{O}_f$ and $a_t \in \mathcal{A}$, assume that

$$
\begin{aligned}
r_f(o_t^f, a_t) &= r(f^{-1}(o_t^f), a_t), \\
\mathcal{P}_f(o_{t+1}^f | o_t^f, a_t) &= \mathcal{P}(f^{-1}(o_{t+1}^f) | f^{-1}(o_t^f), a_t), \quad (2) \\
\rho_f(o_0^f) &= \rho(f^{-1}(o_0^f)).
\end{aligned}
$$

**Explanation.** Assumption 2.2 states that all $\mathcal{M}_f$ share a common underlying MDP $\mathcal{M}$, which is a formal statement of Algorithm 1.

Next, consider an agent interacting with $\mathcal{M}_f$ following the policy $\pi : \mathcal{O} \times \mathcal{A} \mapsto [0, 1]$ to obtain a trajectory

$$
\tau_f = (o_0^f, a_0, r_0^f, o_1^f, a_1, r_1^f, \ldots, o_t^f, a_t, r_t^f, \ldots), \quad (3)
$$

where $o_0^f \sim \rho_f(\cdot)$, $a_t \sim \pi(\cdot | o_t^f)$, $r_t^f = r_f(o_t^f, a_t)$ and $o_{t+1} \sim \mathcal{P}_f(\cdot | o_t^f, a_t)$, we simplify the notation to $\tau_f \sim \pi$.

However, during training, the agent is only allowed to access a subset of all MDPs, which is $\{\mathcal{M}_f | f \in \mathcal{F}_{\text{train}} \subset \mathcal{F}\}$, and then tests its generalization performance across all MDPs. Thus, denote $p_{\text{train}} : \mathcal{F}_{\text{train}} \mapsto [0, 1]$ as the distribution of $\mathcal{F}_{\text{train}}$, the agent's training performance $\eta(\pi)$ and generalization performance $\zeta(\pi)$ can be expressed as

$$
\begin{aligned}
\eta(\pi) &= \mathbb{E}_{f \sim p_{\text{train}}(\cdot), \tau_f \sim \pi} \left[ \sum_{t=0}^{\infty} \gamma^t r_f(o_t^f, a_t) \right], \\
\zeta(\pi) &= \mathbb{E}_{f \sim p(\cdot), \tau_f \sim \pi} \left[ \sum_{t=0}^{\infty} \gamma^t r_f(o_t^f, a_t) \right].
\end{aligned}
\quad (4)
$$

The goal of the agent is to learn a policy $\pi$ that maximizes the generalization performance $\zeta(\pi)$.

## 2.2. Deep Mutual Learning

Deep mutual learning (DML) (Zhang et al., 2018b) is a mutual distillation technique in supervised learning. Unlike the traditional teacher-student distillation strategy, DML aligns the probability distributions of multiple student networks by minimizing the KL divergence loss during training, allowing them to learn from each other. Specifically,

$$
\mathcal{L}_{\text{DML}} = \mathcal{L}_{\text{SL}} + \alpha \mathcal{L}_{\text{KL}}, \quad (5)
$$

where $\mathcal{L}_{\text{SL}}$ and $\mathcal{L}_{\text{KL}}$ represent the supervised learning loss and the KL divergence loss, respectively, $\alpha$ is the weight.

Using DML, the student cohort effectively pools their collective estimate of the next most likely classes. Finding out and matching the other most likely classes for each training instance according to their peers increases each student's posterior entropy, which helps them converge to a more robust representation, leading to better generalization.

## 3. Theoretical Results

In this section, we present the main results of this paper, demonstrating that enhancing the agent's robustness to irrelevant features will improve its generalization performance.

A key issue is that we do not exactly know the probability distribution $p_{\text{train}}$. Note that $\mathcal{F}_{\text{train}}$ is a subset of $\mathcal{F}$, we naturally assume that the probability distribution $p_{\text{train}}$ can be derived from the normalized probability distribution $p$.

**Assumption 3.1.** For any $f \in \mathcal{F}$, assume that

$$
\begin{aligned}
p_{\text{train}}(f) &= \frac{p(f) \cdot \mathbb{I}(f \in \mathcal{F}_{\text{train}})}{Z}, \\
p_{\text{eval}}(f) &= \frac{p(f) \cdot \mathbb{I}(f \in \mathcal{F}_{\text{eval}})}{1 - Z},
\end{aligned}
\quad (6)
$$

where $Z = \int_{\mathcal{F}_{\text{train}}} p(f) \mathrm{d}f$ and $1 - Z$ is the partition function, $\mathcal{F}_{\text{eval}} = \mathcal{F} - \mathcal{F}_{\text{train}}$, $\mathbb{I}(\cdot)$ denotes the indicator function.

An interesting fact is that, for a specific policy $\pi$, if we only consider its interaction with $\mathcal{M}_f$, we can establish a bijection between this policy and a certain underlying policy that directly interacts with $\mathcal{M}$. We now denote it as $\mu_f(\cdot | s_t) = \pi(\cdot | f(s_t))$. By further defining the normalized discounted visitation distribution $d^\mu(s) = (1 - \gamma) \sum_{t=0}^{\infty} \gamma^t \mathbb{P}(s_t = s | \mu)$, we can use this underlying policy $\mu_f$ to replace the training and generalization performance of the policy $\pi$. Specifically,

$$
\begin{aligned}
\eta(\pi) &= \frac{1}{1 - \gamma} \mathbb{E}_{\substack{f \sim p_{\text{train}}(\cdot) \\ s \sim d^{\mu_f}(\cdot) \\ a \sim \mu_f(\cdot | s)}} [r(s, a)], \\
\zeta(\pi) &= \frac{1}{1 - \gamma} \mathbb{E}_{\substack{f \sim p(\cdot) \\ s \sim d^{\mu_f}(\cdot) \\ a \sim \mu_f(\cdot | s)}} [r(s, a)].
\end{aligned}
\quad (7)
$$

We can thus analyze the generalization problem using the underlying policy $\mu_f$. Then, define $L_\pi$ as the first-order approximation of $\eta$ (Schulman, 2015), we can derive the following lower bounds:

**Theorem 3.2** (Training performance lower bound). *Given any two policies, $\tilde{\pi}$ and $\pi$, the following bound holds:*

$$
\eta(\tilde{\pi}) \geq L_\pi(\tilde{\pi}) - \frac{2\gamma \epsilon_{\text{train}}}{(1 - \gamma)^2} \mathbb{E}_{\substack{f \sim p_{\text{train}}(\cdot) \\ s \sim d^{\mu_f}(\cdot)}} [D_{\text{TV}}(\tilde{\mu}_f \| \mu_f)[s]],
$$

(8)

*where $\epsilon_{\text{train}} = \max_{f \in \mathcal{F}_{\text{train}}} \left\{ \max_s \left| \mathbb{E}_{a \sim \tilde{\mu}_f(\cdot | s)} [A^{\mu_f}(s, a)] \right| \right\}$.*

*Proof.* See Appendix A.2. □

**Theorem 3.3** (Generalization performance lower bound).
*Given any two policies, $\tilde{\pi}$ and $\pi$, the following bound holds:*

$$
\begin{aligned}
\zeta(\tilde{\pi}) \geq{}& L_\pi(\tilde{\pi}) - \frac{2r_{\max}(1-Z)}{1-\gamma} \\
&- \frac{2\gamma\epsilon_{\text{train}}}{(1-\gamma)^2} \underset{\substack{f \sim p_{\text{train}}(\cdot) \\ s \sim d^{\mu_f}(\cdot)}}{\mathbb{E}} [D_{\text{TV}}(\tilde{\mu}_f \| \mu_f)[s]] \\
&- \frac{2\delta_{\text{train}}(1-Z)}{1-\gamma} \underset{\substack{f \sim p_{\text{train}}(\cdot) \\ s \sim d^{\tilde{\mu}_f}(\cdot)}}{\mathbb{E}} [D_{\text{TV}}(\tilde{\mu}_f \| \mu_f)[s]] \\
&- \frac{2\delta_{\text{eval}}(1-Z)}{1-\gamma} \underset{\substack{f \sim p_{\text{eval}}(\cdot) \\ s \sim d^{\tilde{\mu}_f}(\cdot)}}{\mathbb{E}} [D_{\text{TV}}(\tilde{\mu}_f \| \mu_f)[s]],
\end{aligned} \tag{9}
$$

*where $r_{\max} = \max_{s,a} |r(s,a)|$, and*

$$
\begin{aligned}
\delta_{\text{train}} &= \max_{f \in \mathcal{F}_{\text{train}}} \left\{ \max_{s,a} |A^{\mu_f}(s,a)| \right\}, \\
\delta_{\text{eval}} &= \max_{f \in \mathcal{F}_{\text{eval}}} \left\{ \max_{s,a} |A^{\mu_f}(s,a)| \right\}.
\end{aligned} \tag{10}
$$

*Proof.* See Appendix A.1. □

**Explanation.** Building on Theorems 3.2 and 3.3, we observe that, in contrast to the lower bound on training performance, the lower bound on generalization performance incorporates three additional terms, scaled by the common coefficient $(1-Z)$. This implies that increasing $Z$ contributes to improved generalization performance, with the special case of $Z = 1$ resulting in alignment between generalization and training performance. Notably, this theoretical insight was also validated in Figure 2 of Cobbe et al. (2020).

However, once the training level is fixed (i.e., $\mathcal{F}_{\text{train}}$), $Z$ is a constant, improving generalization performance requires constraining the following three terms:

$$
\underbrace{\underset{\substack{f \sim p_{\text{train}}(\cdot) \\ s \sim d^{\tilde{\mu}_f}(\cdot)}}{\mathbb{E}} [D_{\text{TV}}(\tilde{\mu}_f \| \mu_f)[s]]}_{\text{denote it as } \mathfrak{D}_1}, \quad \underbrace{\underset{\substack{f \sim p_{\text{eval}}(\cdot) \\ s \sim d^{\tilde{\mu}_f}(\cdot)}}{\mathbb{E}} [D_{\text{TV}}(\tilde{\mu}_f \| \mu_f)[s]]}_{\text{denote it as } \mathfrak{D}_2},
$$

$$\tag{11}$$

and

$$
\underbrace{\underset{\substack{f \sim p_{\text{train}}(\cdot) \\ s \sim d^{\mu_f}(\cdot)}}{\mathbb{E}} [D_{\text{TV}}(\tilde{\mu}_f \| \mu_f)[s]]}_{\text{denote it as } \mathfrak{D}_{\text{train}}}. \tag{12}
$$

During the training process, we can only empirically bound $\mathfrak{D}_{\text{train}}$. Next, we will show that $\mathfrak{D}_{\text{train}}$ is an upper bound of $\mathfrak{D}_1$. Specifically, we propose the following theorem:

**Theorem 3.4.** *Given any two policies, $\tilde{\pi}$ and $\pi$, the following bound holds:*

$$
\mathfrak{D}_1 \leq \left(1 + \frac{2\gamma\sigma_{\text{train}}}{1-\gamma}\right) \mathfrak{D}_{\text{train}}, \tag{13}
$$

*where $\sigma_{\text{train}} = \max_{f \in \mathcal{F}_{\text{train}}} \{\max_s D_{\text{TV}}(\tilde{\mu}_f \| \mu_f)[s]\}$.*

*Proof.* See Appendix A.3. □

Therefore, $\mathfrak{D}_1$ can be bounded by $\mathfrak{D}_{\text{train}}$. As a result, $\mathfrak{D}_2$ becomes crucial for improving generalization performance. Similarly, we can find an upper bound for $\mathfrak{D}_2$.

**Theorem 3.5.** *Given any two policies, $\tilde{\pi}$ and $\pi$, the following bound holds:*

$$
\mathfrak{D}_2 \leq \left(1 + \frac{2\gamma\sigma_{\text{eval}}}{1-\gamma}\right) \underbrace{\underset{\substack{f \sim p_{\text{eval}}(\cdot) \\ s \sim d^{\mu_f}(\cdot)}}{\mathbb{E}} [D_{\text{TV}}(\tilde{\mu}_f \| \mu_f)[s]]}_{\text{denote it as } \mathfrak{D}_{\text{eval}}}, \tag{14}
$$

*where $\sigma_{\text{eval}} = \max_{f \in \mathcal{F}_{\text{eval}}} \{\max_s D_{\text{TV}}(\tilde{\mu}_f \| \mu_f)[s]\}$.*

*Proof.* See Appendix A.4. □

The only problem now is finding the relationship between $\mathfrak{D}_{\text{eval}}$ and $\mathfrak{D}_{\text{train}}$. To achieve this, we would like to first introduce the following definition, which represents the policy robustness to irrelevant features.

**Definition 3.6** ($\mathcal{R}$-robust). We say that the policy $\pi$ is $\mathcal{R}$-robust if it satisfies

$$
\sup_{s \in \mathcal{S}, \tilde{f}, f \in \mathcal{F}} D_{\text{TV}}(\mu_{\tilde{f}} \| \mu_f)[s] = \mathcal{R}. \tag{15}
$$

**Explanation.** This definition demonstrates how the policy $\pi$ is influenced by two different rendering functions, $\tilde{f}$ and $f$, for any given underlying state $s$. If $\mathcal{R} = 0$, it indicates that $D_{\text{TV}}(\mu_{\tilde{f}} \| \mu_f)[s] \equiv 0$, which means that the policy has learned a meta-representation of the observations and is no longer affected by any irrelevant features.

Our intention in this definition is not to derive the tightest possible bound but rather to demonstrate how policy robustness to irrelevant features can contribute to improved generalization. Subsequently, leveraging Definition 3.6, we establish an upper bound for $\mathfrak{D}_{\text{eval}}$.

**Theorem 3.7.** *Given any two policies, $\tilde{\pi}$ and $\pi$, assume that $\tilde{\pi}$ is $\mathcal{R}_{\tilde{\pi}}$-robust, and $\pi$ is $\mathcal{R}_\pi$-robust, then the following bound holds:*

$$
\mathfrak{D}_{\text{eval}} \leq \left(1 + \frac{2\gamma\sigma_{\text{train}}}{1-\gamma}\right) \mathcal{R}_\pi + \mathcal{R}_{\tilde{\pi}} + \mathfrak{D}_{\text{train}}. \tag{16}
$$

*Proof.* See Appendix A.5. □

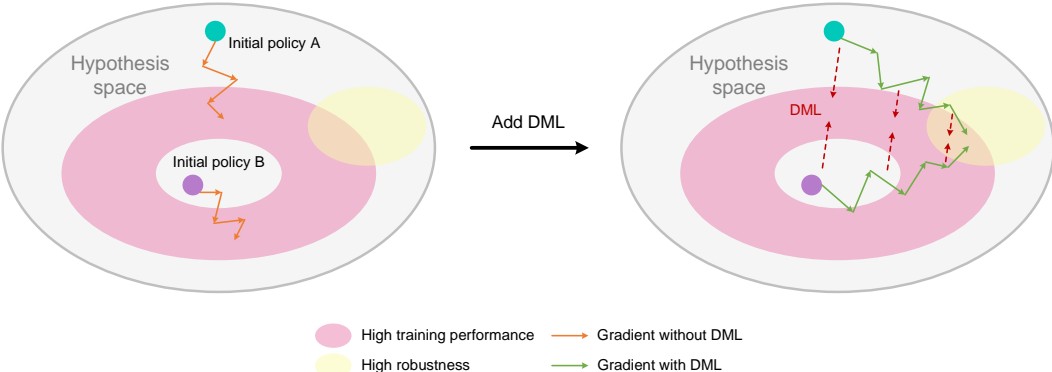

*Figure 2.* Our DML-based technique can drive agents to learn robust representations of noisy observations and gradually reduce the divergence between them, ultimately improving generalization performance.

Altogether, by combining Theorems 3.3, 3.4, 3.5, and 3.7, we can derive the following corollary.

**Corollary 3.8.** *Given any two policies, $\tilde{\pi}$ and $\pi$, the following bound holds:*

$$\zeta(\tilde{\pi}) \geq L_\pi(\tilde{\pi}) - C_{\text{train}}\mathfrak{D}_{\text{train}} - C_\pi \mathcal{R}_\pi - C_{\tilde{\pi}}\mathcal{R}_{\tilde{\pi}} - C, \quad (17)$$

*where*

$$C_{\text{train}} = \frac{2\delta_{\text{train}}(1-Z)}{1-\gamma}\left(1 + \frac{2\gamma\sigma_{\text{train}}}{1-\gamma}\right) + \frac{2\gamma\epsilon_{\text{train}}}{(1-\gamma)^2}$$
$$+ \frac{2\delta_{\text{eval}}(1-Z)}{1-\gamma}\left(1 + \frac{2\gamma\sigma_{\text{eval}}}{1-\gamma}\right),$$
$$C_\pi = \frac{2\delta_{\text{eval}}(1-Z)}{1-\gamma}\left(1 + \frac{2\gamma\sigma_{\text{eval}}}{1-\gamma}\right)\left(1 + \frac{2\gamma\sigma_{\text{train}}}{1-\gamma}\right),$$
$$C_{\tilde{\pi}} = \frac{2\delta_{\text{eval}}(1-Z)}{1-\gamma}\left(1 + \frac{2\gamma\sigma_{\text{eval}}}{1-\gamma}\right), \quad C = \frac{2r_{\max}(1-Z)}{1-\gamma}.$$
$$(18)$$

**Explanation.** This represents our central theoretical result, demonstrating that enhancing generalization performance requires not only minimizing $\mathfrak{D}_{\text{train}}$ during training but also improving policy robustness to irrelevant features, specifically by reducing $\mathcal{R}_\pi$ and $\mathcal{R}_{\tilde{\pi}}$. Furthermore, we emphasize that these results rely solely on the mild Assumptions 2.1, 2.2, and 3.1. Consequently, this constitutes a novel contribution that is broadly applicable to a wide range of algorithms.

## 4. Central Hypothesis

Despite the theoretical advancements, in typical generalization settings, both the underlying Markov Decision Process (MDP) and the rendering function remain unknown. In this section, we propose that deep mutual learning (DML) (Zhang et al., 2018b) can be leveraged to enhance policy robustness against irrelevant features in high-dimensional observations, thereby improving generalization performance. This hypothesis is further illustrated in Figure 2.

> **The Meta-Representation Hypothesis**
>
> We propose a hypothesis that deep mutual learning (DML) technique can help agents learn meta-representations of high-dimensional observations, thus improving generalization performance.

The figure illustrates two randomly initialized policies independently trained using reinforcement learning algorithms. In this case, since the training samples only include a portion of the MDPs, the policies are likely to overfit to irrelevant features and fail to converge to a robust hypothesis space.

Introducing the DML loss into the training process of two policies (denoted as policy A and policy B) facilitates mutual learning, which can mitigate overfitting to irrelevant features. Due to the random initialization of policies A and B, they generate different training samples. The DML loss encourages both policies to make consistent decisions on the same observations. As a result, any irrelevant features learned by policy A are likely to degrade the performance of policy B (see Appendix B for further explanation), and vice versa. As training progresses, DML will drive both policies to learn more meaningful and useful representations, gradually reducing the divergence between them (right of the Figure 2). Ideally, we hypothesize that both policies will converge to meta-representations that capture the essential aspects of high-dimensional observations as time grows.

An intriguing analogy for our hypothesis is the process of truth emergence. Typically, each scholar offers their unique perspective, but for it to be widely accepted, it must garner consensus from peers within the field, or even from the broader academic community. We can draw a parallel between DML and the peer review process: when a particular viewpoint is accepted by the majority, it is more likely to reflect an objective truth—though, of course, this does not preclude the possibility that everyone could be mistaken, as

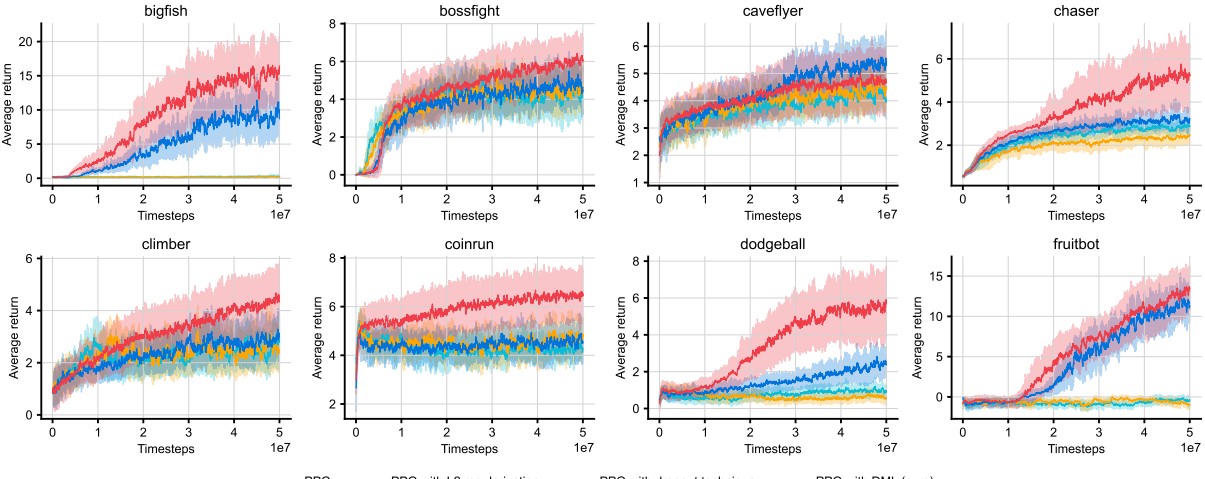

*Figure 3.* Generalization performance from 500 levels in Procgen benchmark with different methods. The mean and standard deviation are shown across 3 seeds. Our proposed DML-based method gains significant improvement compared with the baseline algorithm.

seen during the era when geocentrism was widely endorsed. On a deeper level, our hypothesis aligns with the philosophical concept of convergent realism (Laudan, 1981; Kelly & Glymour, 1989; Huh et al., 2024), which posits that science progresses towards an objective truth.

## 5. Experiments

### 5.1. Implementation Details

We use Procgen (Cobbe et al., 2019; 2020) as the experimental benchmark for testing generalization performance. Procgen is a suite of 16 procedurally generated game-like environments designed to benchmark both sample efficiency and generalization in reinforcement learning, and it has been widely used to test the generalization performance of various reinforcement learning algorithms (Wang et al., 2020; Raileanu & Fergus, 2021; Raileanu et al., 2021; Lyle et al., 2022; Rahman & Xue, 2023; Jesson & Jiang, 2024).

We employ the Proximal Policy Optimization (PPO) (Schulman et al., 2017; Cobbe et al., 2020) algorithm as our baseline, as PPO is one of the most widely used model-free reinforcement learning algorithms. Specifically, given a parameterized policy $\pi_\theta$ ($\theta$ represents the parameters), the objective of $\pi_\theta$ is to maximize

$$\mathbb{E}_{(o_t, a_t) \sim \pi_{\theta_{\text{old}}}} \left\{ \min \left[ r_t(\theta) \cdot \hat{A}(o_t, a_t), \tilde{r}_t(\theta) \cdot \hat{A}(o_t, a_t) \right] \right\}, \tag{19}$$

where $\hat{A}$ is the advantage estimate, and

$$r_t(\theta) = \frac{\pi_\theta(a_t|o_t)}{\pi_{\theta_{\text{old}}}(a_t|o_t)}, \quad \tilde{r}_t(\theta) = \text{clip}\left(r_t(\theta), 1 - \epsilon, 1 + \epsilon\right), \tag{20}$$

with $\pi_{\theta_{\text{old}}}$ and $\pi_\theta$ being the old policy and the current policy.

---

**Algorithm 2** PPO with DML

1: **Initialize:** Two agents $\pi_1, \pi_2$, PPO algorithm $\mathcal{A}$, KL divergence weight $\alpha$
2: **while** training **do**
3:     **for** $i = 1, 2$ **do**
4:         Collect training data: $\mathcal{D}_i \sim \pi_i$
5:         Compute RL loss: $\mathcal{L}_{\text{RL}}^{(i)} \leftarrow \mathcal{A}(\mathcal{D}_i)$
6:         Compute KL loss: $\mathcal{L}_{\text{KL}}^{(i)} \leftarrow D_{\text{KL}}(\pi_{3-i} \| \pi_i)$
7:         Compute DML loss: $\mathcal{L}_{\text{DML}}^{(i)} \leftarrow \mathcal{L}_{\text{RL}}^{(i)} + \alpha \mathcal{L}_{\text{KL}}^{(i)}$
8:     **end for**
9:     Compute total loss: $\mathcal{L} \leftarrow \frac{1}{2} \left( \mathcal{L}_{\text{DML}}^{(1)} + \mathcal{L}_{\text{DML}}^{(2)} \right)$
10:    Optimize $\mathcal{L}$ using gradient descent algorithm
11: **end while**

---

We randomly initialize two agents to interact with the environment and collect data separately. Similar to the DML loss (5) used in supervised learning, we also introduce an additional KL divergence loss term, which is

$$\mathcal{L}_{\text{DML}} = \mathcal{L}_{\text{RL}} + \alpha \mathcal{L}_{\text{KL}}, \tag{21}$$

where $\mathcal{L}_{\text{RL}}$ is the reinforcement learning loss and $\mathcal{L}_{\text{KL}}$ is the KL divergence loss, $\alpha$ is the weight. And then we optimize the total loss of both agents, which is the average of their DML losses, as shown in Algorithm 2.

Finally, we do not claim to achieve state-of-the-art (SOTA) performance, but rather to verify that the DML technique indeed helps agents learn more robust representations from high-dimensional observations and leads to consistent improvements in generalization performance, providing empirical support for our central theory and hypothesis.

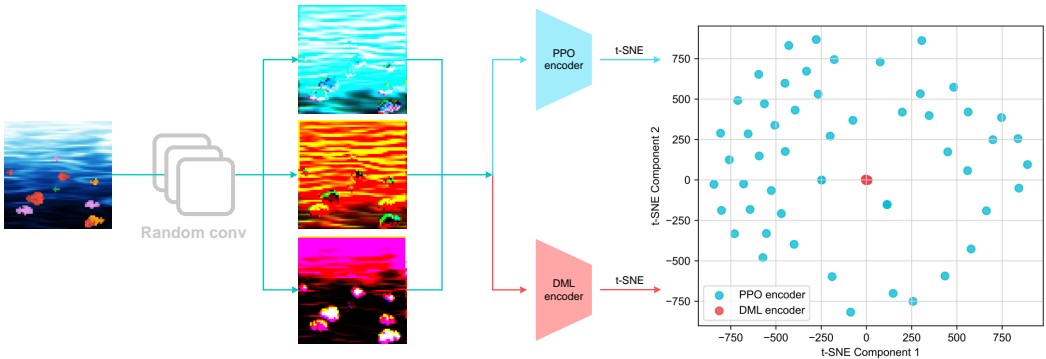

*Figure 4.* To test the robustness of the trained policy, we obfuscate the agent's observations using convolutional layers randomly initialized with a standard Gaussian distribution. If the agent has indeed learned to ignore irrelevant features from noisy observations, it should exhibit better robustness to such obfuscations. Notably, the feature extraction of the PPO encoder enhanced by DML is highly ***stable and focused*** (red points), whereas the features extracted by the original PPO encoder are significantly dispersed (blue points).

*Table 1.* We input each current frame into 100 randomly initialized convolutional layers and calculate the average changes in KL divergence according to Section 5.3. The table presents the mean and standard deviation of the recorded data over 100 consecutive interaction steps. In this context, lower mean and standard deviation indicate a more robust policy.

| Algorithm\Environment | bigfish | bossfight | caveflyer | chaser | climber | coinrun | dodgeball | fruitbot |
|---|---|---|---|---|---|---|---|---|
| PPO | $6.15 \pm 1.58$ | $8.19 \pm 0.96$ | $8.60 \pm 0.73$ | $14.40 \pm 1.19$ | $0.62 \pm 0.48$ | $1.70 \pm 0.35$ | $0.09 \pm 0.05$ | $6.41 \pm 1.25$ |
| PPO with DML | $3.91 \pm 0.58$ | $0.32 \pm 0.20$ | $1.38 \pm 0.35$ | $4.66 \pm 0.70$ | $0.09 \pm 0.06$ | $1.57 \pm 0.41$ | $1.29 \pm 0.16$ | $1.22 \pm 0.63$ |

| Algorithm\Environment | heist | jumper | leaper | maze | miner | ninja | plunder | starpilot |
|---|---|---|---|---|---|---|---|---|
| PPO | $1.38 \pm 0.19$ | $10.92 \pm 1.80$ | $4.94 \pm 1.57$ | $5.79 \pm 1.10$ | $12.44 \pm 3.28$ | $5.91 \pm 1.30$ | $4.61 \pm 1.00$ | $3.72 \pm 0.93$ |
| PPO with DML | $0.05 \pm 0.03$ | $0.75 \pm 0.29$ | $2.33 \pm 1.07$ | $1.44 \pm 0.33$ | $2.08 \pm 0.86$ | $2.30 \pm 1.07$ | $2.44 \pm 0.36$ | $2.88 \pm 0.75$ |

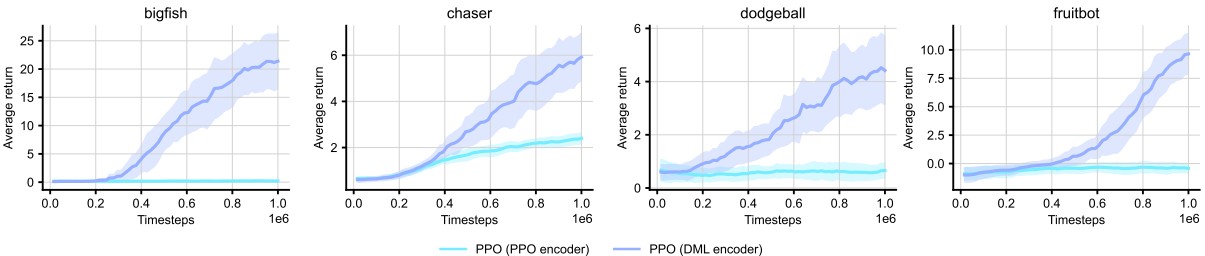

*Figure 5.* Generalization performance of retraining policies using the frozen encoders obtained from the PPO baseline and our method.

## 5.2. Empirical Results

We compare the generalization performance of our approach against the PPO baseline on the Procgen benchmark, under the hard-level settings (Cobbe et al., 2020). The results are illustrated in Figure 3. It can be observed that DML technique indeed leads to consistent improvements in generalization performance across all environments. Notably, for the bigfish, dodgeball, and fruitbot environments, we have observed significant improvements. Moreover, the experimental results for all 16 environments in Procgen benchmark, including training performance and generalization performance, can be found in Appendix C.

A natural concern arises: how can we determine whether DML improves generalization performance by enhancing the policy robustness against irrelevant features, or simply due to the additional information sharing between these two agents during training (each agent receives additional information than it would from training alone)? To answer this question, we conducted robustness testing on the trained policies in Section 5.3 and added an ablation study in Section 5.4 to verify our theory and hypothesis.

## 5.3. Robustness Testing

To further verify that our method has indeed learned more robust policies, we design a novel approach to test policy robustness against irrelevant features, as shown in Figure 4. For each current frame, we input it into multiple convolutional layers randomly initialized with a standard Gaussian

distribution, and then compute the average KL divergence of the policy before and after the perturbation by these random convolutional layers. This allows us to effectively test the robustness of the trained policies without changing the underlying semantics. The results can be seen from Table 1. We can see that the average changes in KL divergence of our method is lower than the PPO baseline across almost all environments, with a smaller standard deviation, providing strong empirical support for our central hypothesis.

Moreover, we employ t-SNE to visualize the agent's encoding of high-dimensional observations in the bigfish environment, as shown in Figure 4. Each scatter point represents a low-dimensional embedding of a vector obtained by passing the current pixel input through these random convolutional layers, and then fed into the agent's encoder. It can be observed that the scatter points of our method are more tightly clustered, indicating a more robust representation of high-dimensional noisy observations, which serves as further strong evidence for our hypothesis.

**5.4. Ablation Study**

To verify that the generalization performance of the agent benefits from more robust policies, we designed additional ablation experiments. Specifically, we used the frozen encoders obtained from the PPO baseline and our method to retrain the policies, the results are shown in Figure 5. Since the policy obtained from our method is more robust to irrelevant features (as demonstrated in Section 5.3), the encoder learns a better representation of the high-dimensional observations. Therefore, based on our theoretical results, retraining policies using the frozen encoders obtained from our method should have better generalization performance. We can see that the generalization performance in Figure 5 strongly supports our theoretical results.

In summary, Section 5.2 validates the effectiveness of DML technique for generalization, Section 5.3 verifies our central hypothesis, and Section 5.4 confirms our theoretical results.

## 6. Related Work

The generalization of deep reinforcement learning has been widely studied, and previous work has pointed out the overfitting problem in deep reinforcement learning (Rajeswaran et al., 2017; Zhang et al., 2018a; Justesen et al., 2018; Packer et al., 2018; Song et al., 2019; Cobbe et al., 2019; Grigsby & Qi, 2020; Cobbe et al., 2020; Yuan et al., 2024).

A natural approach to avoid the overfitting problem in deep reinforcement learning is to apply regularization techniques originally developed for supervised learning such as dropout (Srivastava et al., 2014; Farebrother et al., 2018; Igl et al., 2019), data augmentation (Laskin et al., 2020; Kostrikov et al., 2020; Zhang & Guo, 2021; Raileanu et al., 2021;

Ma et al., 2022), domain randomization (Tobin et al., 2017; Yue et al., 2019; Slaoui et al., 2019; Mehta et al., 2020), or network randomization technique (Lee et al., 2019).

On the other hand, in order to improve sample efficiency, previous studies encouraged the policy network and value network to share parameters (Schulman et al., 2017; Huang et al., 2022). However, recent works have explored the idea of decoupling the two and proposed additional distillation strategies (Cobbe et al., 2021; Raileanu & Fergus, 2021; Moon et al., 2022). In particular, Raileanu & Fergus (2021) demonstrated that more information is needed to accurately estimate the value function, which can lead to overfitting.

## 7. Conclusion

In this paper, we provide a novel theoretical framework to explain the generalization problem in deep reinforcement learning. We also hypothesize that the DML technique facilitates meta-representation learning. Strong empirical results support our central theory and hypothesis, demonstrating that our approach can improve the generalization performance of RL systems by enhancing robustness against irrelevant features. Our work provides valuable insights into the development of more adaptable and robust RL systems capable of generalizing across diverse domains.

## 8. Discussion

A key insight from our work is the process of extracting patterns from empirical observations, a powerful abstraction ability that is central to human cognition. This raises a fundamental question: if human perception is based on electrical and chemical signals in the brain, then how can we infer the true nature of the world?

Our approach offers a potential answer through the concept of cognitive alignment (Falandays & Smaldino, 2022). By encouraging agents to make consistent decisions based on the same observations, our method fosters a process akin to cognitive alignment, which has been fundamental in human societal development. For instance, in voting, the majority rule is employed because decisions supported by the majority are perceived as more reliable. Similarly, our method facilitates cognitive alignment between agents, enabling them to converge on objective truths despite noisy or irrelevant features in their observations.

Over time, the cognitive alignment between agents encourages the convergence of their individual representations toward a more accurate understanding of the environment. This process mirrors the philosophical notion presented in Plato's *Allegory of the Cave* (Cohen, 2006), where individuals break free from the constraints of their limited perceptions to grasp the true nature of reality.

## Impact Statement

This paper presents work whose goal is to advance the field of Machine Learning. There are many potential societal consequences of our work, none which we feel must be specifically highlighted here.

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

## A. Proofs

Let's start with some useful lemmas.

**Lemma A.1** (Performance difference). *Let $\mu_f(\cdot|s_t) = \pi(\cdot|f(s_t))$ and $\tilde{\mu}_f(\cdot|s_t) = \tilde{\pi}(\cdot|f(s_t))$, define training and generalization performance as*

$$\eta(\pi) = \frac{1}{1-\gamma} \mathop{\mathbb{E}}_{\substack{f \sim p_{\text{train}}(\cdot) \\ s \sim d^{\mu_f}(\cdot) \\ a \sim \mu_f(\cdot|s)}} [r(s,a)], \quad \zeta(\pi) = \frac{1}{1-\gamma} \mathop{\mathbb{E}}_{\substack{f \sim p(\cdot) \\ s \sim d^{\mu_f}(\cdot) \\ a \sim \mu_f(\cdot|s)}} [r(s,a)]. \tag{22}$$

*Then the differences in training and generalization performance can be expressed as*

$$\eta(\tilde{\pi}) - \eta(\pi) = \frac{1}{1-\gamma} \mathop{\mathbb{E}}_{\substack{f \sim p_{\text{train}}(\cdot) \\ s \sim d^{\tilde{\mu}_f}(\cdot) \\ a \sim \tilde{\mu}_f(\cdot|s)}} [A^{\mu_f}(s,a)], \quad \zeta(\tilde{\pi}) - \zeta(\pi) = \frac{1}{1-\gamma} \mathop{\mathbb{E}}_{\substack{f \sim p(\cdot) \\ s \sim d^{\tilde{\mu}_f}(\cdot) \\ a \sim \tilde{\mu}_f(\cdot|s)}} [A^{\mu_f}(s,a)]. \tag{23}$$

*Proof.* This result can be directly derived from Kakade & Langford (2002). $\qquad\square$

**Lemma A.2.** *The divergence between two normalized discounted visitation distribution, $\|d^{\tilde{\mu}} - d^{\mu}\|_1$, is bounded by an average divergence of $\tilde{\mu}$ and $\mu$:*

$$\|d^{\tilde{\mu}} - d^{\mu}\|_1 \le \frac{\gamma}{1-\gamma} \mathop{\mathbb{E}}_{s \sim d^{\mu}(\cdot)} [\|\tilde{\mu} - \mu\|_1] = \frac{2\gamma}{1-\gamma} \mathop{\mathbb{E}}_{s \sim d^{\mu}(\cdot)} [D_{\text{TV}}(\tilde{\mu}\|\mu)[s]], \tag{24}$$

*where $D_{\text{TV}}(\tilde{\mu}\|\mu)[s] = \frac{1}{2}\sum_{a\in\mathcal{A}}|\tilde{\mu}(a|s) - \mu(a|s)|$ represents the Total Variation (TV) distance.*

*Proof.* See Achiam et al. (2017). $\qquad\square$

**Lemma A.3.** *Given any state $s \in \mathcal{S}$, any two policies $\tilde{\mu}$ and $\mu$, the average advantage, $\mathbb{E}_{a\sim\tilde{\mu}(\cdot|s)}[A^{\mu}(s,a)]$, is bounded by*

$$\left|\mathbb{E}_{a\sim\tilde{\mu}(\cdot|s)}[A^{\mu}(s,a)]\right| \le 2D_{\text{TV}}(\tilde{\mu}\|\mu)[s] \cdot \max_a |A^{\mu}(s,a)|. \tag{25}$$

*Proof.* Note that

$$\begin{aligned} \mathbb{E}_{a\sim\mu(\cdot|s)}[A^{\mu}(s,a)] &= \mathbb{E}_{a\sim\mu(\cdot|s)}[Q^{\mu}(s,a) - V^{\mu}(s)] \\ &= \mathbb{E}_{a\sim\mu(\cdot|s)}[Q^{\mu}(s,a)] - V^{\mu}(s) \\ &= V^{\mu}(s) - V^{\mu}(s) \\ &= 0, \end{aligned} \tag{26}$$

thus,

$$\begin{aligned} \left|\mathbb{E}_{a\sim\tilde{\mu}(\cdot|s)}[A^{\mu}(s,a)]\right| &= \left|\mathbb{E}_{a\sim\tilde{\mu}(\cdot|s)}[A^{\mu}(s,a)] - \mathbb{E}_{a\sim\mu(\cdot|s)}[A^{\mu}(s,a)]\right| \\ &\le \|\tilde{\mu}(a|s) - \mu(a|s)\|_1 \cdot \|A^{\mu}(s,a)\|_{\infty} \\ &= 2D_{\text{TV}}(\tilde{\mu}\|\mu)[s] \cdot \max_a |A^{\mu}(s,a)|. \end{aligned} \tag{27}$$

This is a widely used trick (Schulman, 2015; Zhuang et al., 2023; Gan et al., 2024). $\qquad\square$

In addition, using the above lemmas, the following corollary can be obtained, which will be repeatedly used in our proof.

**Corollary A.4.** *Given any two policies, $\tilde{\mu}$ and $\mu$, the following bound holds:*

$$\left| \mathop{\mathbb{E}}_{\substack{s \sim d^{\tilde{\mu}}(\cdot) \\ a \sim \tilde{\mu}(\cdot|s)}} [A^{\mu}(s,a)] - \mathop{\mathbb{E}}_{\substack{s \sim d^{\mu}(\cdot) \\ a \sim \tilde{\mu}(\cdot|s)}} [A^{\mu}(s,a)] \right| \le \frac{2\epsilon\gamma}{1-\gamma} \mathop{\mathbb{E}}_{s \sim d^{\mu}(\cdot)} [D_{\text{TV}}(\tilde{\mu}\|\mu)[s]], \tag{28}$$

*where $\epsilon = \max_s \left|\mathbb{E}_{a\sim\tilde{\mu}(\cdot|s)}[A^{\mu}(s,a)]\right|$.*

*Proof.* We rewrite the expectation as

$$\left| \mathop{\mathbb{E}}_{\substack{s \sim d^{\tilde{\mu}}(\cdot) \\ a \sim \tilde{\mu}(\cdot|s)}} [A^{\mu}(s,a)] - \mathop{\mathbb{E}}_{\substack{s \sim d^{\mu}(\cdot) \\ a \sim \tilde{\mu}(\cdot|s)}} [A^{\mu}(s,a)] \right| = \left| \mathop{\mathbb{E}}_{s \sim d^{\tilde{\mu}}(\cdot)} \left\{ \mathop{\mathbb{E}}_{a \sim \tilde{\mu}(\cdot|s)} [A^{\mu}(s,a)] \right\} - \mathop{\mathbb{E}}_{s \sim d^{\mu}(\cdot)} \left\{ \mathop{\mathbb{E}}_{a \sim \tilde{\mu}(\cdot|s)} [A^{\mu}(s,a)] \right\} \right|, \quad (29)$$

where the expectation $\mathbb{E}_{a \sim \tilde{\mu}(\cdot|s)} [A^{\mu}(s,a)]$ is a function of $s$, then

$$\left| \mathop{\mathbb{E}}_{s \sim d^{\tilde{\mu}}(\cdot)} \left\{ \mathop{\mathbb{E}}_{a \sim \tilde{\mu}(\cdot|s)} [A^{\mu}(s,a)] \right\} - \mathop{\mathbb{E}}_{s \sim d^{\mu}(\cdot)} \left\{ \mathop{\mathbb{E}}_{a \sim \tilde{\mu}(\cdot|s)} [A^{\mu}(s,a)] \right\} \right| \leq \left\| d^{\tilde{\mu}} - d^{\mu} \right\|_1 \cdot \left\| \mathop{\mathbb{E}}_{a \sim \tilde{\mu}(\cdot|s)} [A^{\mu}(s,a)] \right\|_{\infty}. \quad (30)$$

Next, according to Lemma A.2, we have

$$\left\| d^{\tilde{\mu}} - d^{\mu} \right\|_1 \cdot \left\| \mathop{\mathbb{E}}_{a \sim \tilde{\mu}(\cdot|s)} [A^{\mu}(s,a)] \right\|_{\infty} = \epsilon \left\| d^{\tilde{\mu}} - d^{\mu} \right\|_1 \leq \frac{2\epsilon\gamma}{1-\gamma} \mathop{\mathbb{E}}_{s \sim d^{\mu}(\cdot)} [D_{\text{TV}}(\tilde{\mu}\|\mu)[s]], \quad (31)$$

concluding the proof. $\qquad\square$

### A.1. Proof of Theorem 3.3

**Theorem 3.3.** *Given any two policies, $\tilde{\pi}$ and $\pi$, the following bound holds:*

$$\zeta(\tilde{\pi}) \geq L_{\pi}(\tilde{\pi}) - \frac{2r_{\max}(1-Z)}{1-\gamma} - \frac{2\gamma\epsilon_{\text{train}}}{(1-\gamma)^2} \mathop{\mathbb{E}}_{\substack{f \sim p_{\text{train}}(\cdot) \\ s \sim d^{\mu_f}(\cdot)}} [D_{\text{TV}}(\tilde{\mu}_f\|\mu_f)[s]]$$

$$- \frac{2\delta_{\text{train}}(1-Z)}{1-\gamma} \mathop{\mathbb{E}}_{\substack{f \sim p_{\text{train}}(\cdot) \\ s \sim d^{\tilde{\mu}_f}(\cdot)}} [D_{\text{TV}}(\tilde{\mu}_f\|\mu_f)[s]] - \frac{2\delta_{\text{eval}}(1-Z)}{1-\gamma} \mathop{\mathbb{E}}_{\substack{f \sim p_{\text{eval}}(\cdot) \\ s \sim d^{\tilde{\mu}_f}(\cdot)}} [D_{\text{TV}}(\tilde{\mu}_f\|\mu_f)[s]]. \quad (32)$$

*Proof.* Let's start with the first-order approximation of the training performance (Schulman, 2015), denote it as

$$L_{\pi}(\tilde{\pi}) = \eta(\pi) + \frac{1}{1-\gamma} \mathop{\mathbb{E}}_{\substack{f \sim p_{\text{train}}(\cdot) \\ s \sim d^{\mu_f}(\cdot) \\ a \sim \tilde{\mu}_f(\cdot|s)}} [A^{\mu_f}(s,a)]. \quad (33)$$

Then, we are trying to bound the difference between $\zeta(\tilde{\pi})$ and $L_{\pi}(\tilde{\pi})$, according to Lemma A.1, that is,

$$|\zeta(\tilde{\pi}) - L_{\pi}(\tilde{\pi})|$$

$$= \left| \zeta(\pi) - \eta(\pi) + \frac{1}{1-\gamma} \mathop{\mathbb{E}}_{\substack{f \sim p(\cdot) \\ s \sim d^{\tilde{\mu}_f}(\cdot) \\ a \sim \tilde{\mu}_f(\cdot|s)}} [A^{\mu_f}(s,a)] - \frac{1}{1-\gamma} \mathop{\mathbb{E}}_{\substack{f \sim p_{\text{train}}(\cdot) \\ s \sim d^{\mu_f}(\cdot) \\ a \sim \tilde{\mu}_f(\cdot|s)}} [A^{\mu_f}(s,a)] \right|$$

$$= \frac{1}{1-\gamma} \left| \mathop{\mathbb{E}}_{\substack{f \sim p(\cdot) \\ s \sim d^{\mu_f}(\cdot) \\ a \sim \mu_f(\cdot|s)}} [r(s,a)] - \mathop{\mathbb{E}}_{\substack{f \sim p_{\text{train}}(\cdot) \\ s \sim d^{\mu_f}(\cdot) \\ a \sim \mu_f(\cdot|s)}} [r(s,a)] + \mathop{\mathbb{E}}_{\substack{f \sim p(\cdot) \\ s \sim d^{\tilde{\mu}_f}(\cdot) \\ a \sim \tilde{\mu}_f(\cdot|s)}} [A^{\mu_f}(s,a)] - \mathop{\mathbb{E}}_{\substack{f \sim p_{\text{train}}(\cdot) \\ s \sim d^{\mu_f}(\cdot) \\ a \sim \tilde{\mu}_f(\cdot|s)}} [A^{\mu_f}(s,a)] \right| \quad (34)$$

$$\leq \frac{1}{1-\gamma} \left\{ \left| \mathop{\mathbb{E}}_{\substack{f \sim p(\cdot) \\ s \sim d^{\mu_f}(\cdot) \\ a \sim \mu_f(\cdot|s)}} [r(s,a)] - \mathop{\mathbb{E}}_{\substack{f \sim p_{\text{train}}(\cdot) \\ s \sim d^{\mu_f}(\cdot) \\ a \sim \mu_f(\cdot|s)}} [r(s,a)] \right| + \left| \mathop{\mathbb{E}}_{\substack{f \sim p(\cdot) \\ s \sim d^{\tilde{\mu}_f}(\cdot) \\ a \sim \tilde{\mu}_f(\cdot|s)}} [A^{\mu_f}(s,a)] - \mathop{\mathbb{E}}_{\substack{f \sim p_{\text{train}}(\cdot) \\ s \sim d^{\mu_f}(\cdot) \\ a \sim \tilde{\mu}_f(\cdot|s)}} [A^{\mu_f}(s,a)] \right| \right\}.$$

We can bound these two terms separately. Simplifying the notation, denote $g(f) = \mathbb{E}_{s\sim d^{\mu_f}(\cdot),a\sim\mu_f(\cdot|s)}[r(s,a)]$, we can thus rewrite the first term as

$$\left| \mathop{\mathbb{E}}_{\substack{f\sim p(\cdot)\\ s\sim d^{\mu_f}(\cdot)\\ a\sim\mu_f(\cdot|s)}} [r(s,a)] - \mathop{\mathbb{E}}_{\substack{f\sim p_{\text{train}}(\cdot)\\ s\sim d^{\mu_f}(\cdot)\\ a\sim\mu_f(\cdot|s)}} [r(s,a)] \right| = \left| \mathop{\mathbb{E}}_{f\sim p(\cdot)} [g(f)] - \mathop{\mathbb{E}}_{f\sim p_{\text{train}}(\cdot)} [g(f)] \right|, \tag{35}$$

then

$$\left| \mathop{\mathbb{E}}_{f\sim p(\cdot)} [g(f)] - \mathop{\mathbb{E}}_{f\sim p_{\text{train}}(\cdot)} [g(f)] \right| = \left| \int_{\mathcal{F}} p(f)\cdot g(f)\mathrm{d}f - \int_{\mathcal{F}_{\text{train}}} p_{\text{train}}(f)\cdot g(f)\mathrm{d}f \right|. \tag{36}$$

Next, according to Assumption 3.1,

$$
\begin{aligned}
& \left| \int_{\mathcal{F}} p(f)\cdot g(f)\mathrm{d}f - \int_{\mathcal{F}_{\text{train}}} p_{\text{train}}(f)\cdot g(f)\mathrm{d}f \right| \\
&= \left| \int_{\mathcal{F}} p(f)\cdot g(f)\mathrm{d}f - \int_{\mathcal{F}_{\text{train}}} \frac{p(f)}{Z}\cdot g(f)\mathrm{d}f \right| \\
&= \left| \int_{\mathcal{F}_{\text{train}}} p(f)\cdot g(f)\mathrm{d}f - \int_{\mathcal{F}_{\text{train}}} \frac{p(f)}{Z}\cdot g(f)\mathrm{d}f + \int_{\mathcal{F}-\mathcal{F}_{\text{train}}} p(f)\cdot g(f)\mathrm{d}f \right| \\
&= \left| \int_{\mathcal{F}_{\text{train}}} \frac{Z-1}{Z} p(f)\cdot g(f)\mathrm{d}f + \int_{\mathcal{F}-\mathcal{F}_{\text{train}}} p(f)\cdot g(f)\mathrm{d}f \right|,
\end{aligned}
\tag{37}
$$

where $Z = \int_{\mathcal{F}_{\text{train}}} p(f)\mathrm{d}f \le 1$, thus,

$$
\begin{aligned}
& \left| \int_{\mathcal{F}_{\text{train}}} \frac{Z-1}{Z} p(f)\cdot g(f)\mathrm{d}f + \int_{\mathcal{F}-\mathcal{F}_{\text{train}}} p(f)\cdot g(f)\mathrm{d}f \right| \\
&\le \left| \int_{\mathcal{F}_{\text{train}}} \frac{Z-1}{Z} p(f)\cdot g(f)\mathrm{d}f \right| + \left| \int_{\mathcal{F}-\mathcal{F}_{\text{train}}} p(f)\cdot g(f)\mathrm{d}f \right| \\
&\le \frac{1-Z}{Z} \left| \int_{\mathcal{F}_{\text{train}}} p(f)\cdot g(f)\mathrm{d}f \right| + \left| \int_{\mathcal{F}-\mathcal{F}_{\text{train}}} p(f)\cdot g(f)\mathrm{d}f \right|.
\end{aligned}
\tag{38}
$$

Meanwhile,

$$
\begin{aligned}
|g(f)| &= \left| \mathop{\mathbb{E}}_{\substack{s\sim d^{\mu_f}(\cdot)\\ a\sim\mu_f(\cdot|s)}} [r(s,a)] \right| = \left| \sum_{s\in\mathcal{S}}(1-\gamma)\sum_{t=0}^{\infty}\gamma^t \mathbb{P}(s_t=s|\mu_f)\sum_{a\in\mathcal{A}}\mu_f(a|s)\cdot r(s,a) \right| \\
&\le (1-\gamma)\sum_{t=0}^{\infty}\sum_{s\in\mathcal{S}}\mathbb{P}(s_t=s|\mu_f)\sum_{a\in\mathcal{A}}\mu_f(a|s)\cdot\gamma^t |r(s,a)| \\
&\le (1-\gamma)\sum_{t=0}^{\infty}\gamma^t r_{\max} = r_{\max},
\end{aligned}
\tag{39}
$$

where $r_{\max} = \max_{s,a} |r(s,a)|$, then we can bound the first term as

$$
\left| \mathop{\mathbb{E}}_{\substack{f\sim p(\cdot)\\ s\sim d^{\mu_f}(\cdot)\\ a\sim \mu_f(\cdot|s)}} [r(s,a)] - \mathop{\mathbb{E}}_{\substack{f\sim p_{\text{train}}(\cdot)\\ s\sim d^{\mu_f}(\cdot)\\ a\sim \mu_f(\cdot|s)}} [r(s,a)] \right| \le \frac{1-Z}{Z} \left| \int_{\mathcal{F}_{\text{train}}} p(f)\cdot g(f)\mathrm{d}f \right| + \left| \int_{\mathcal{F}-\mathcal{F}_{\text{train}}} p(f)\cdot g(f)\mathrm{d}f \right|
$$

$$
\le \frac{1-Z}{Z} \int_{\mathcal{F}_{\text{train}}} p(f)\cdot |g(f)|\,\mathrm{d}f + \int_{\mathcal{F}-\mathcal{F}_{\text{train}}} p(f)\cdot |g(f)|\,\mathrm{d}f \tag{40}
$$

$$
\le \frac{(1-Z)r_{\max}}{Z} \int_{\mathcal{F}_{\text{train}}} p(f)\mathrm{d}f + r_{\max} \int_{\mathcal{F}-\mathcal{F}_{\text{train}}} p(f)\mathrm{d}f
$$

$$
= \frac{(1-Z)r_{\max}}{Z} \cdot Z + r_{\max}\cdot(1-Z) = 2r_{\max}(1-Z).
$$

Now we are trying to bound the second term, which can be expressed as

$$
\left| \mathop{\mathbb{E}}_{\substack{f\sim p(\cdot)\\ s\sim d^{\tilde{\mu}_f}(\cdot)\\ a\sim \tilde{\mu}_f(\cdot|s)}} [A^{\mu_f}(s,a)] - \mathop{\mathbb{E}}_{\substack{f\sim p_{\text{train}}(\cdot)\\ s\sim d^{\mu_f}(\cdot)\\ a\sim \tilde{\mu}_f(\cdot|s)}} [A^{\mu_f}(s,a)] \right|
$$

$$
= \left| \mathop{\mathbb{E}}_{\substack{f\sim p(\cdot)\\ s\sim d^{\tilde{\mu}_f}(\cdot)\\ a\sim \tilde{\mu}_f(\cdot|s)}} [A^{\mu_f}(s,a)] - \mathop{\mathbb{E}}_{\substack{f\sim p_{\text{train}}(\cdot)\\ s\sim d^{\tilde{\mu}_f}(\cdot)\\ a\sim \tilde{\mu}_f(\cdot|s)}} [A^{\mu_f}(s,a)] + \mathop{\mathbb{E}}_{\substack{f\sim p_{\text{train}}(\cdot)\\ s\sim d^{\tilde{\mu}_f}(\cdot)\\ a\sim \tilde{\mu}_f(\cdot|s)}} [A^{\mu_f}(s,a)] - \mathop{\mathbb{E}}_{\substack{f\sim p_{\text{train}}(\cdot)\\ s\sim d^{\mu_f}(\cdot)\\ a\sim \tilde{\mu}_f(\cdot|s)}} [A^{\mu_f}(s,a)] \right| \tag{41}
$$

$$
\le \underbrace{\left| \mathop{\mathbb{E}}_{\substack{f\sim p(\cdot)\\ s\sim d^{\tilde{\mu}_f}(\cdot)\\ a\sim \tilde{\mu}_f(\cdot|s)}} [A^{\mu_f}(s,a)] - \mathop{\mathbb{E}}_{\substack{f\sim p_{\text{train}}(\cdot)\\ s\sim d^{\tilde{\mu}_f}(\cdot)\\ a\sim \tilde{\mu}_f(\cdot|s)}} [A^{\mu_f}(s,a)] \right|}_{\text{denote as } \Phi} + \underbrace{\left| \mathop{\mathbb{E}}_{\substack{f\sim p_{\text{train}}(\cdot)\\ s\sim d^{\tilde{\mu}_f}(\cdot)\\ a\sim \tilde{\mu}_f(\cdot|s)}} [A^{\mu_f}(s,a)] - \mathop{\mathbb{E}}_{\substack{f\sim p_{\text{train}}(\cdot)\\ s\sim d^{\mu_f}(\cdot)\\ a\sim \tilde{\mu}_f(\cdot|s)}} [A^{\mu_f}(s,a)] \right|}_{\text{denote as } \Psi}.
$$

Using Corollary A.4, $\Psi$ can be bounded by

$$
\Psi = \left| \mathop{\mathbb{E}}_{f\sim p_{\text{train}}(\cdot)} \left\{ \mathop{\mathbb{E}}_{\substack{s\sim d^{\tilde{\mu}_f}(\cdot)\\ a\sim \tilde{\mu}_f(\cdot|s)}} [A^{\mu_f}(s,a)] - \mathop{\mathbb{E}}_{\substack{s\sim d^{\mu_f}(\cdot)\\ a\sim \tilde{\mu}_f(\cdot|s)}} [A^{\mu_f}(s,a)] \right\} \right|
$$

$$
\le \mathop{\mathbb{E}}_{f\sim p_{\text{train}}(\cdot)} \left\{ \left| \mathop{\mathbb{E}}_{\substack{s\sim d^{\tilde{\mu}_f}(\cdot)\\ a\sim \tilde{\mu}_f(\cdot|s)}} [A^{\mu_f}(s,a)] - \mathop{\mathbb{E}}_{\substack{s\sim d^{\mu_f}(\cdot)\\ a\sim \tilde{\mu}_f(\cdot|s)}} [A^{\mu_f}(s,a)] \right| \right\} \tag{42}
$$

$$
\le \mathop{\mathbb{E}}_{f\sim p_{\text{train}}(\cdot)} \left\{ \frac{2\epsilon\gamma}{1-\gamma} \mathop{\mathbb{E}}_{s\sim d^{\mu_f}(\cdot)} [D_{\text{TV}}(\tilde{\mu}_f\|\mu_f)[s]] \right\},
$$

where $\epsilon = \max_s \left| \mathbb{E}_{a\sim \tilde{\mu}_f(\cdot|s)} [A^{\mu_f}(s,a)] \right|$, denote $\epsilon_{\text{train}} = \max_{f\in\mathcal{F}_{\text{train}}} \left\{ \max_s \left| \mathbb{E}_{a\sim \tilde{\mu}_f(\cdot|s)} [A^{\mu_f}(s,a)] \right| \right\}$, we obtain

$$
\Psi \le \frac{2\gamma\epsilon_{\text{train}}}{1-\gamma} \mathop{\mathbb{E}}_{\substack{f\sim p_{\text{train}}(\cdot)\\ s\sim d^{\mu_f}(\cdot)}} [D_{\text{TV}}(\tilde{\mu}_f\|\mu_f)[s]]. \tag{43}
$$

Next, with a little abuse of notation $g(f)$, denote

$$
g(f) = \mathop{\mathbb{E}}_{\substack{s\sim d^{\tilde{\mu}_f}(\cdot)\\ a\sim \tilde{\mu}_f(\cdot|s)}} [A^{\mu_f}(s,a)], \tag{44}
$$

we can rewrite $\Phi$ as

$$\Phi = \left| \mathop{\mathbb{E}}_{f \sim p(\cdot)} [g(f)] - \mathop{\mathbb{E}}_{f \sim p_{\mathrm{train}}(\cdot)} [g(f)] \right|, \tag{45}$$

then, similar to (36), (37), (38) and (40),

$$\Phi \leq \frac{1-Z}{Z} \int_{\mathcal{F}_{\mathrm{train}}} p(f) \cdot |g(f)| \, \mathrm{d}f + \int_{\mathcal{F} - \mathcal{F}_{\mathrm{train}}} p(f) \cdot |g(f)| \, \mathrm{d}f. \tag{46}$$

According to Lemma A.3, we can bound $g(f)$, which can be expressed as

$$g(f) = \mathop{\mathbb{E}}_{\substack{s \sim d^{\tilde{\mu}_f}(\cdot) \\ a \sim \tilde{\mu}_f(\cdot|s)}} [A^{\mu_f}(s,a)] = \mathop{\mathbb{E}}_{s \sim d^{\tilde{\mu}_f}(\cdot)} \left\{ \mathop{\mathbb{E}}_{a \sim \tilde{\mu}_f(\cdot|s)} [A^{\mu_f}(s,a)] \right\}, \tag{47}$$

thus,

$$|g(f)| \leq \mathop{\mathbb{E}}_{s \sim d^{\tilde{\mu}_f}(\cdot)} \left\{ \left| \mathop{\mathbb{E}}_{a \sim \tilde{\mu}_f(\cdot|s)} [A^{\mu_f}(s,a)] \right| \right\} \leq \mathop{\mathbb{E}}_{s \sim d^{\tilde{\mu}_f}(\cdot)} \left\{ 2D_{\mathrm{TV}}(\tilde{\mu}_f \| \mu_f)[s] \cdot \max_a |A^{\mu_f}(s,a)| \right\}. \tag{48}$$

Denote $\delta = \max_{s,a} |A^{\mu_f}(s,a)|$, then we have

$$|g(f)| \leq 2\delta \mathop{\mathbb{E}}_{s \sim d^{\tilde{\mu}_f}(\cdot)} [D_{\mathrm{TV}}(\tilde{\mu}_f \| \mu_f)[s]], \tag{49}$$

which means that

$$\Phi \leq \frac{1-Z}{Z} \int_{\mathcal{F}_{\mathrm{train}}} p(f) \cdot |g(f)| \, \mathrm{d}f + \int_{\mathcal{F} - \mathcal{F}_{\mathrm{train}}} p(f) \cdot |g(f)| \, \mathrm{d}f$$

$$\leq \frac{2\delta_{\mathrm{train}}(1-Z)}{Z} \int_{\mathcal{F}_{\mathrm{train}}} p(f) \cdot \mathop{\mathbb{E}}_{s \sim d^{\tilde{\mu}_f}(\cdot)} [D_{\mathrm{TV}}(\tilde{\mu}_f \| \mu_f)[s]] \, \mathrm{d}f + 2\delta_{\mathrm{eval}} \int_{\mathcal{F} - \mathcal{F}_{\mathrm{train}}} p(f) \cdot \mathop{\mathbb{E}}_{s \sim d^{\tilde{\mu}_f}(\cdot)} [D_{\mathrm{TV}}(\tilde{\mu}_f \| \mu_f)[s]] \, \mathrm{d}f$$

$$= 2\delta_{\mathrm{train}}(1-Z) \int_{\mathcal{F}_{\mathrm{train}}} \frac{p(f)}{Z} \cdot \mathop{\mathbb{E}}_{s \sim d^{\tilde{\mu}_f}(\cdot)} [D_{\mathrm{TV}}(\tilde{\mu}_f \| \mu_f)[s]] \, \mathrm{d}f + 2\delta_{\mathrm{eval}}(1-Z) \int_{\mathcal{F} - \mathcal{F}_{\mathrm{train}}} \frac{p(f)}{1-Z} \cdot \mathop{\mathbb{E}}_{s \sim d^{\tilde{\mu}_f}(\cdot)} [D_{\mathrm{TV}}(\tilde{\mu}_f \| \mu_f)[s]] \, \mathrm{d}f$$

$$= 2\delta_{\mathrm{train}}(1-Z) \mathop{\mathbb{E}}_{\substack{f \sim p_{\mathrm{train}}(\cdot) \\ s \sim d^{\tilde{\mu}_f}(\cdot)}} [D_{\mathrm{TV}}(\tilde{\mu}_f \| \mu_f)[s]] + 2\delta_{\mathrm{eval}}(1-Z) \mathop{\mathbb{E}}_{\substack{f \sim p_{\mathrm{eval}}(\cdot) \\ s \sim d^{\tilde{\mu}_f}(\cdot)}} [D_{\mathrm{TV}}(\tilde{\mu}_f \| \mu_f)[s]], \tag{50}$$

where $\delta_{\mathrm{train}} = \max_{f \in \mathcal{F}_{\mathrm{train}}} \{\max_{s,a} |A^{\mu_f}(s,a)|\}$ and $\delta_{\mathrm{eval}} = \max_{f \in \mathcal{F}_{\mathrm{eval}}} \{\max_{s,a} |A^{\mu_f}(s,a)|\}$.

Finally, combining (34), (40), (41), (43), and (50), we have

$$|\zeta(\tilde{\pi}) - L_\pi(\tilde{\pi})| \leq \frac{2r_{\max}(1-Z)}{1-\gamma} + \frac{2\gamma\epsilon_{\mathrm{train}}}{(1-\gamma)^2} \mathop{\mathbb{E}}_{\substack{f \sim p_{\mathrm{train}}(\cdot) \\ s \sim d^{\mu_f}(\cdot)}} [D_{\mathrm{TV}}(\tilde{\mu}_f \| \mu_f)[s]]$$

$$+ \frac{2\delta_{\mathrm{train}}(1-Z)}{1-\gamma} \mathop{\mathbb{E}}_{\substack{f \sim p_{\mathrm{train}}(\cdot) \\ s \sim d^{\tilde{\mu}_f}(\cdot)}} [D_{\mathrm{TV}}(\tilde{\mu}_f \| \mu_f)[s]] + \frac{2\delta_{\mathrm{eval}}(1-Z)}{1-\gamma} \mathop{\mathbb{E}}_{\substack{f \sim p_{\mathrm{eval}}(\cdot) \\ s \sim d^{\tilde{\mu}_f}(\cdot)}} [D_{\mathrm{TV}}(\tilde{\mu}_f \| \mu_f)[s]], \tag{51}$$

thus, the generalization performance lower bound is

$$\zeta(\tilde{\pi}) \geq L_\pi(\tilde{\pi}) - \frac{2r_{\max}(1-Z)}{1-\gamma} - \frac{2\gamma\epsilon_{\mathrm{train}}}{(1-\gamma)^2} \mathop{\mathbb{E}}_{\substack{f \sim p_{\mathrm{train}}(\cdot) \\ s \sim d^{\mu_f}(\cdot)}} [D_{\mathrm{TV}}(\tilde{\mu}_f \| \mu_f)[s]]$$

$$- \frac{2\delta_{\mathrm{train}}(1-Z)}{1-\gamma} \mathop{\mathbb{E}}_{\substack{f \sim p_{\mathrm{train}}(\cdot) \\ s \sim d^{\tilde{\mu}_f}(\cdot)}} [D_{\mathrm{TV}}(\tilde{\mu}_f \| \mu_f)[s]] - \frac{2\delta_{\mathrm{eval}}(1-Z)}{1-\gamma} \mathop{\mathbb{E}}_{\substack{f \sim p_{\mathrm{eval}}(\cdot) \\ s \sim d^{\tilde{\mu}_f}(\cdot)}} [D_{\mathrm{TV}}(\tilde{\mu}_f \| \mu_f)[s]], \tag{52}$$

concluding the proof. $\qquad\qquad \square$

## A.2. Proof of Theorem 3.2

**Theorem 3.2.** *Given any two policies, $\tilde{\pi}$ and $\pi$, the following bound holds:*

$$\eta(\tilde{\pi}) \geq L_\pi(\tilde{\pi}) - \frac{2\gamma\epsilon_{\text{train}}}{(1-\gamma)^2} \mathop{\mathbb{E}}_{\substack{f \sim p_{\text{train}}(\cdot) \\ s \sim d^{\mu_f}(\cdot)}} \left[ D_{\text{TV}}(\tilde{\mu}_f \| \mu_f)[s] \right]. \tag{53}$$

*Proof.* Since

$$|\eta(\tilde{\pi}) - L_\pi(\tilde{\pi})| = \frac{1}{1-\gamma} \left| \mathop{\mathbb{E}}_{\substack{f \sim p_{\text{train}}(\cdot) \\ s \sim d^{\tilde{\mu}_f}(\cdot) \\ a \sim \tilde{\mu}_f(\cdot|s)}} [A^{\mu_f}(s,a)] - \mathop{\mathbb{E}}_{\substack{f \sim p_{\text{train}}(\cdot) \\ s \sim d^{\mu_f}(\cdot) \\ a \sim \tilde{\mu}_f(\cdot|s)}} [A^{\mu_f}(s,a)] \right| = \frac{\Psi}{1-\gamma}$$

$$\leq \frac{2\gamma\epsilon_{\text{train}}}{(1-\gamma)^2} \mathop{\mathbb{E}}_{\substack{f \sim p_{\text{train}}(\cdot) \\ s \sim d^{\mu_f}(\cdot)}} \left[ D_{\text{TV}}(\tilde{\mu}_f \| \mu_f)[s] \right], \tag{54}$$

thus,

$$\eta(\tilde{\pi}) \geq L_\pi(\tilde{\pi}) - \frac{2\gamma\epsilon_{\text{train}}}{(1-\gamma)^2} \mathop{\mathbb{E}}_{\substack{f \sim p_{\text{train}}(\cdot) \\ s \sim d^{\mu_f}(\cdot)}} \left[ D_{\text{TV}}(\tilde{\mu}_f \| \mu_f)[s] \right], \tag{55}$$

concluding the proof. $\square$

## A.3. Proof of Theorem 3.4

**Theorem 3.4.** *Given any two policies, $\tilde{\pi}$ and $\pi$, the following bound holds:*

$$\mathfrak{D}_1 \leq \left( 1 + \frac{2\gamma\sigma_{\text{train}}}{1-\gamma} \right) \mathfrak{D}_{\text{train}}, \tag{56}$$

*where $\sigma_{\text{train}} = \max_{f \in \mathcal{F}_{\text{train}}} \{ \max_s D_{\text{TV}}(\tilde{\mu}_f \| \mu_f)[s] \}$.*

*Proof.* According to Lemma A.2, we have

$$|\mathfrak{D}_1 - \mathfrak{D}_{\text{train}}| = \left| \mathop{\mathbb{E}}_{\substack{f \sim p_{\text{train}}(\cdot) \\ s \sim d^{\tilde{\mu}_f}(\cdot)}} \left[ D_{\text{TV}}(\tilde{\mu}_f \| \mu_f)[s] \right] - \mathop{\mathbb{E}}_{\substack{f \sim p_{\text{train}}(\cdot) \\ s \sim d^{\mu_f}(\cdot)}} \left[ D_{\text{TV}}(\tilde{\mu}_f \| \mu_f)[s] \right] \right|$$

$$= \left| \mathop{\mathbb{E}}_{f \sim p_{\text{train}}(\cdot)} \left\{ \mathop{\mathbb{E}}_{s \sim d^{\tilde{\mu}_f}(\cdot)} \left[ D_{\text{TV}}(\tilde{\mu}_f \| \mu_f)[s] \right] - \mathop{\mathbb{E}}_{s \sim d^{\mu_f}(\cdot)} \left[ D_{\text{TV}}(\tilde{\mu}_f \| \mu_f)[s] \right] \right\} \right|$$

$$\leq \mathop{\mathbb{E}}_{f \sim p_{\text{train}}(\cdot)} \left\{ \left| \mathop{\mathbb{E}}_{s \sim d^{\tilde{\mu}_f}(\cdot)} \left[ D_{\text{TV}}(\tilde{\mu}_f \| \mu_f)[s] \right] - \mathop{\mathbb{E}}_{s \sim d^{\mu_f}(\cdot)} \left[ D_{\text{TV}}(\tilde{\mu}_f \| \mu_f)[s] \right] \right| \right\} \tag{57}$$

$$\leq \mathop{\mathbb{E}}_{f \sim p_{\text{train}}(\cdot)} \left\{ \left\| d^{\tilde{\mu}_f} - d^{\mu_f} \right\|_1 \cdot \left\| D_{\text{TV}}(\tilde{\mu}_f \| \mu_f)[s] \right\|_\infty \right\}$$

$$\leq \mathop{\mathbb{E}}_{f \sim p_{\text{train}}(\cdot)} \left\{ \frac{2\gamma}{1-\gamma} \mathop{\mathbb{E}}_{s \sim d^{\mu_f}(\cdot)} \left[ D_{\text{TV}}(\tilde{\mu}_f \| \mu_f)[s] \right] \cdot \max_s D_{\text{TV}}(\tilde{\mu}_f \| \mu_f)[s] \right\}$$

$$\leq \frac{2\gamma\sigma_{\text{train}}}{1-\gamma} \mathop{\mathbb{E}}_{\substack{f \sim p_{\text{train}}(\cdot) \\ s \sim d^{\mu_f}(\cdot)}} \left[ D_{\text{TV}}(\tilde{\mu}_f \| \mu_f)[s] \right] = \frac{2\gamma\sigma_{\text{train}}}{1-\gamma} \cdot \mathfrak{D}_{\text{train}},$$

as a result,

$$\mathfrak{D}_1 \leq \left( 1 + \frac{2\gamma\sigma_{\text{train}}}{1-\gamma} \right) \mathfrak{D}_{\text{train}}, \tag{58}$$

concluding the proof. $\square$

### A.4. Proof of Theorem 3.5

**Theorem 3.5.** *Given any two policies, $\tilde{\pi}$ and $\pi$, the following bound holds:*

$$\mathfrak{D}_2 \leq \left(1 + \frac{2\gamma\sigma_{\text{eval}}}{1-\gamma}\right) \underbrace{\mathbb{E}_{\substack{f \sim p_{\text{eval}}(\cdot) \\ s \sim d^{\mu_f}(\cdot)}}[D_{\text{TV}}(\tilde{\mu}_f \| \mu_f)[s]]}_{\text{denote it as } \mathfrak{D}_{\text{eval}}}, \tag{59}$$

*where $\sigma_{\text{eval}} = \max_{f \in \mathcal{F}_{\text{eval}}} \{\max_s D_{\text{TV}}(\tilde{\mu}_f \| \mu_f)[s]\}$.*

*Proof.* Similar to the proof of Theorem 3.4, using Lemma A.2 again, we have

$$
\begin{aligned}
|\mathfrak{D}_2 - \mathfrak{D}_{\text{eval}}| &= \left| \mathbb{E}_{\substack{f \sim p_{\text{eval}}(\cdot) \\ s \sim d^{\tilde{\mu}_f}(\cdot)}}[D_{\text{TV}}(\tilde{\mu}_f \| \mu_f)[s]] - \mathbb{E}_{\substack{f \sim p_{\text{eval}}(\cdot) \\ s \sim d^{\mu_f}(\cdot)}}[D_{\text{TV}}(\tilde{\mu}_f \| \mu_f)[s]] \right| \\
&= \left| \mathbb{E}_{f \sim p_{\text{eval}}(\cdot)} \left\{ \mathbb{E}_{s \sim d^{\tilde{\mu}_f}(\cdot)}[D_{\text{TV}}(\tilde{\mu}_f \| \mu_f)[s]] - \mathbb{E}_{s \sim d^{\mu_f}(\cdot)}[D_{\text{TV}}(\tilde{\mu}_f \| \mu_f)[s]] \right\} \right| \\
&\leq \mathbb{E}_{f \sim p_{\text{eval}}(\cdot)} \left\{ \left| \mathbb{E}_{s \sim d^{\tilde{\mu}_f}(\cdot)}[D_{\text{TV}}(\tilde{\mu}_f \| \mu_f)[s]] - \mathbb{E}_{s \sim d^{\mu_f}(\cdot)}[D_{\text{TV}}(\tilde{\mu}_f \| \mu_f)[s]] \right| \right\} \\
&\leq \mathbb{E}_{f \sim p_{\text{eval}}(\cdot)} \left\{ \left\| d^{\tilde{\mu}_f} - d^{\mu_f} \right\|_1 \cdot \| D_{\text{TV}}(\tilde{\mu}_f \| \mu_f)[s] \|_\infty \right\} \\
&\leq \mathbb{E}_{f \sim p_{\text{eval}}(\cdot)} \left\{ \frac{2\gamma}{1-\gamma} \mathbb{E}_{s \sim d^{\mu_f}(\cdot)}[D_{\text{TV}}(\tilde{\mu}_f \| \mu_f)[s]] \cdot \max_s D_{\text{TV}}(\tilde{\mu}_f \| \mu_f)[s] \right\} \\
&\leq \frac{2\gamma\sigma_{\text{eval}}}{1-\gamma} \mathbb{E}_{\substack{f \sim p_{\text{eval}}(\cdot) \\ s \sim d^{\mu_f}(\cdot)}}[D_{\text{TV}}(\tilde{\mu}_f \| \mu_f)[s]] = \frac{2\gamma\sigma_{\text{eval}}}{1-\gamma} \cdot \mathfrak{D}_{\text{eval}},
\end{aligned}
\tag{60}
$$

as a result,

$$\mathfrak{D}_2 \leq \left(1 + \frac{2\gamma\sigma_{\text{eval}}}{1-\gamma}\right) \mathfrak{D}_{\text{eval}}, \tag{61}$$

concluding the proof. $\square$

### A.5. Proof of Theorem 3.7

**Theorem 3.7.** *Given any two policies, $\tilde{\pi}$ and $\pi$, assume that $\tilde{\pi}$ is $\mathcal{R}_{\tilde{\pi}}$-robust, and $\pi$ is $\mathcal{R}_\pi$-robust, then the following bound holds:*

$$\mathfrak{D}_{\text{eval}} \leq \left(1 + \frac{2\gamma\sigma_{\text{train}}}{1-\gamma}\right) \mathcal{R}_\pi + \mathcal{R}_{\tilde{\pi}} + \mathfrak{D}_{\text{train}}. \tag{62}$$

*Proof.* Let's first rewrite $\mathfrak{D}_{\text{eval}}$ as

$$\mathfrak{D}_{\text{eval}} = \mathbb{E}_{\substack{\tilde{f} \sim p_{\text{eval}}(\cdot) \\ s \sim d^{\mu_{\tilde{f}}}(\cdot)}} \left[ D_{\text{TV}}(\tilde{\mu}_{\tilde{f}} \| \mu_{\tilde{f}})[s] \right]. \tag{63}$$

For another $f \in \mathcal{F}_{\text{train}}$, by repeatedly using the triangle inequality of the TV distance, we have

$$
\begin{aligned}
\mathfrak{D}_{\text{eval}} &= \mathbb{E}_{\substack{\tilde{f} \sim p_{\text{eval}}(\cdot) \\ s \sim d^{\mu_{\tilde{f}}}(\cdot)}} \left[ D_{\text{TV}}(\tilde{\mu}_{\tilde{f}} \| \mu_{\tilde{f}})[s] \right] \\
&\leq \mathbb{E}_{\substack{\tilde{f} \sim p_{\text{eval}}(\cdot) \\ s \sim d^{\mu_{\tilde{f}}}(\cdot)}} \left[ D_{\text{TV}}(\tilde{\mu}_{\tilde{f}} \| \tilde{\mu}_f)[s] + D_{\text{TV}}(\tilde{\mu}_f \| \mu_f)[s] + D_{\text{TV}}(\mu_f \| \mu_{\tilde{f}})[s] \right] \\
&= \mathbb{E}_{\substack{\tilde{f} \sim p_{\text{eval}}(\cdot) \\ s \sim d^{\mu_{\tilde{f}}}(\cdot)}} \left[ D_{\text{TV}}(\tilde{\mu}_{\tilde{f}} \| \tilde{\mu}_f)[s] \right] + \mathbb{E}_{\substack{\tilde{f} \sim p_{\text{eval}}(\cdot) \\ s \sim d^{\mu_{\tilde{f}}}(\cdot)}} \left[ D_{\text{TV}}(\tilde{\mu}_f \| \mu_f)[s] \right] + \mathbb{E}_{\substack{\tilde{f} \sim p_{\text{eval}}(\cdot) \\ s \sim d^{\mu_{\tilde{f}}}(\cdot)}} \left[ D_{\text{TV}}(\mu_f \| \mu_{\tilde{f}})[s] \right],
\end{aligned}
\tag{64}
$$

taking the expectation of both sides of the inequality with respect to $f \sim p_{\text{train}}(\cdot)$, we obtain

$$\mathbb{E}_{f \sim p_{\text{train}}(\cdot)} [\mathfrak{D}_{\text{eval}}] \leq \mathbb{E}_{\substack{f \sim p_{\text{train}}(\cdot) \\ \tilde{f} \sim p_{\text{eval}}(\cdot) \\ s \sim d^{\mu_{\tilde{f}}}(\cdot)}} \left[ D_{\text{TV}}(\tilde{\mu}_{\tilde{f}} \| \tilde{\mu}_f)[s] \right] + \mathbb{E}_{\substack{f \sim p_{\text{train}}(\cdot) \\ \tilde{f} \sim p_{\text{eval}}(\cdot) \\ s \sim d^{\mu_{\tilde{f}}}(\cdot)}} [D_{\text{TV}}(\tilde{\mu}_f \| \mu_f)[s]] + \mathbb{E}_{\substack{f \sim p_{\text{train}}(\cdot) \\ \tilde{f} \sim p_{\text{eval}}(\cdot) \\ s \sim d^{\mu_{\tilde{f}}}(\cdot)}} \left[ D_{\text{TV}}(\mu_f \| \mu_{\tilde{f}})[s] \right]. \quad (65)$$

Since $\mathfrak{D}_{\text{eval}}$ is independent of $f$, it becomes a constant after taking the expectation, which is

$$\mathfrak{D}_{\text{eval}} \leq \mathbb{E}_{\substack{f \sim p_{\text{train}}(\cdot) \\ \tilde{f} \sim p_{\text{eval}}(\cdot) \\ s \sim d^{\mu_{\tilde{f}}}(\cdot)}} \left[ D_{\text{TV}}(\tilde{\mu}_{\tilde{f}} \| \tilde{\mu}_f)[s] \right] + \mathbb{E}_{\substack{f \sim p_{\text{train}}(\cdot) \\ \tilde{f} \sim p_{\text{eval}}(\cdot) \\ s \sim d^{\mu_{\tilde{f}}}(\cdot)}} [D_{\text{TV}}(\tilde{\mu}_f \| \mu_f)[s]] + \mathbb{E}_{\substack{f \sim p_{\text{train}}(\cdot) \\ \tilde{f} \sim p_{\text{eval}}(\cdot) \\ s \sim d^{\mu_{\tilde{f}}}(\cdot)}} \left[ D_{\text{TV}}(\mu_f \| \mu_{\tilde{f}})[s] \right]. \quad (66)$$

Note that $\tilde{\pi}$ is $\mathcal{R}_{\tilde{\pi}}$-robust, and $\pi$ is $\mathcal{R}_{\pi}$-robust, we can thus bound the first term:

$$\mathbb{E}_{\substack{f \sim p_{\text{train}}(\cdot) \\ \tilde{f} \sim p_{\text{eval}}(\cdot) \\ s \sim d^{\mu_{\tilde{f}}}(\cdot)}} \left[ D_{\text{TV}}(\tilde{\mu}_{\tilde{f}} \| \tilde{\mu}_f)[s] \right] = \mathbb{E}_{\substack{f \sim p_{\text{train}}(\cdot) \\ \tilde{f} \sim p_{\text{eval}}(\cdot)}} \left[ \sum_{s \in \mathcal{S}} d^{\mu_{\tilde{f}}}(s) \cdot D_{\text{TV}}(\tilde{\mu}_{\tilde{f}} \| \tilde{\mu}_f)[s] \right]$$

$$\leq \mathbb{E}_{\substack{f \sim p_{\text{train}}(\cdot) \\ \tilde{f} \sim p_{\text{eval}}(\cdot)}} \left[ \sum_{s \in \mathcal{S}} d^{\mu_{\tilde{f}}}(s) \cdot \mathcal{R}_{\tilde{\pi}} \right] = \mathcal{R}_{\tilde{\pi}} \mathbb{E}_{\substack{f \sim p_{\text{train}}(\cdot) \\ \tilde{f} \sim p_{\text{eval}}(\cdot)}} \left[ \sum_{s \in \mathcal{S}} d^{\mu_{\tilde{f}}}(s) \right] = \mathcal{R}_{\tilde{\pi}}. \quad (67)$$

Similarly, we can bound the third term:

$$\mathbb{E}_{\substack{f \sim p_{\text{train}}(\cdot) \\ \tilde{f} \sim p_{\text{eval}}(\cdot) \\ s \sim d^{\mu_{\tilde{f}}}(\cdot)}} \left[ D_{\text{TV}}(\mu_{\tilde{f}} \| \mu_f)[s] \right] = \mathbb{E}_{\substack{f \sim p_{\text{train}}(\cdot) \\ \tilde{f} \sim p_{\text{eval}}(\cdot)}} \left[ \sum_{s \in \mathcal{S}} d^{\mu_{\tilde{f}}}(s) \cdot D_{\text{TV}}(\mu_{\tilde{f}} \| \mu_f)[s] \right]$$

$$\leq \mathbb{E}_{\substack{f \sim p_{\text{train}}(\cdot) \\ \tilde{f} \sim p_{\text{eval}}(\cdot)}} \left[ \sum_{s \in \mathcal{S}} d^{\mu_{\tilde{f}}}(s) \cdot \mathcal{R}_{\pi} \right] = \mathcal{R}_{\pi} \mathbb{E}_{\substack{f \sim p_{\text{train}}(\cdot) \\ \tilde{f} \sim p_{\text{eval}}(\cdot)}} \left[ \sum_{s \in \mathcal{S}} d^{\mu_{\tilde{f}}}(s) \right] = \mathcal{R}_{\pi}. \quad (68)$$

Next, we are trying to bound the second term, which is similar to $\mathfrak{D}_{\text{train}}$. Note that $\mathfrak{D}_{\text{train}}$ is independent of $\tilde{f}$, we can thus rewrite it as

$$\mathfrak{D}_{\text{train}} = \mathbb{E}_{\substack{f \sim p_{\text{train}}(\cdot) \\ s \sim d^{\mu_f}(\cdot)}} [D_{\text{TV}}(\tilde{\mu}_f \| \mu_f)[s]] = \mathbb{E}_{\substack{f \sim p_{\text{train}}(\cdot) \\ \tilde{f} \sim p_{\text{eval}}(\cdot) \\ s \sim d^{\mu_f}(\cdot)}} [D_{\text{TV}}(\tilde{\mu}_f \| \mu_f)[s]], \quad (69)$$

then

$$\left| \mathbb{E}_{\substack{f \sim p_{\text{train}}(\cdot) \\ \tilde{f} \sim p_{\text{eval}}(\cdot) \\ s \sim d^{\mu_{\tilde{f}}}(\cdot)}} [D_{\text{TV}}(\tilde{\mu}_f \| \mu_f)[s]] - \mathfrak{D}_{\text{train}} \right|$$

$$= \left| \mathbb{E}_{\substack{f \sim p_{\text{train}}(\cdot) \\ \tilde{f} \sim p_{\text{eval}}(\cdot) \\ s \sim d^{\mu_{\tilde{f}}}(\cdot)}} [D_{\text{TV}}(\tilde{\mu}_f \| \mu_f)[s]] - \mathbb{E}_{\substack{f \sim p_{\text{train}}(\cdot) \\ \tilde{f} \sim p_{\text{eval}}(\cdot) \\ s \sim d^{\mu_f}(\cdot)}} [D_{\text{TV}}(\tilde{\mu}_f \| \mu_f)[s]] \right| \quad (70)$$

$$= \left| \int_{\mathcal{F}_{\text{train}}} p_{\text{train}}(f) \int_{\mathcal{F}_{\text{eval}}} p_{\text{eval}}(\tilde{f}) \left\{ \mathbb{E}_{s \sim d^{\mu_{\tilde{f}}}(\cdot)} [D_{\text{TV}}(\tilde{\mu}_f \| \mu_f)[s]] - \mathbb{E}_{s \sim d^{\mu_f}(\cdot)} [D_{\text{TV}}(\tilde{\mu}_f \| \mu_f)[s]] \right\} \mathrm{d}\tilde{f} \mathrm{d}f \right|$$

$$\leq \int_{\mathcal{F}_{\text{train}}} p_{\text{train}}(f) \int_{\mathcal{F}_{\text{eval}}} p_{\text{eval}}(\tilde{f}) \left\{ \left| \mathbb{E}_{s \sim d^{\mu_{\tilde{f}}}(\cdot)} [D_{\text{TV}}(\tilde{\mu}_f \| \mu_f)[s]] - \mathbb{E}_{s \sim d^{\mu_f}(\cdot)} [D_{\text{TV}}(\tilde{\mu}_f \| \mu_f)[s]] \right| \right\} \mathrm{d}\tilde{f} \mathrm{d}f.$$

Note that,

$$\left| \mathop{\mathbb{E}}_{s \sim d^{\mu_{\tilde{f}}}(\cdot)} [D_{\mathrm{TV}}(\tilde{\mu}_f \| \mu_f)[s]] - \mathop{\mathbb{E}}_{s \sim d^{\mu_f}(\cdot)} [D_{\mathrm{TV}}(\tilde{\mu}_f \| \mu_f)[s]] \right| \leq \|d^{\mu_{\tilde{f}}} - d^{\mu_f}\|_1 \cdot \|D_{\mathrm{TV}}(\tilde{\mu}_f \| \mu_f)[s]\|_\infty. \tag{71}$$

According to Lemma A.2,

$$\|d^{\mu_{\tilde{f}}} - d^{\mu_f}\|_1 \leq \frac{2\gamma}{1-\gamma} \mathop{\mathbb{E}}_{s \sim d^{\mu_f}(\cdot)} \left[ D_{\mathrm{TV}}(\mu_{\tilde{f}} \| \mu_f)[s] \right], \tag{72}$$

$\pi$ is $\mathcal{R}_\pi$-robust, so,

$$\|d^{\mu_{\tilde{f}}} - d^{\mu_f}\|_1 \leq \frac{2\gamma}{1-\gamma} \mathop{\mathbb{E}}_{s \sim d^{\mu_f}(\cdot)} \left[ D_{\mathrm{TV}}(\mu_{\tilde{f}} \| \mu_f)[s] \right] = \frac{2\gamma}{1-\gamma} \sum_{s \in \mathcal{S}} d^{\mu_f}(s) \cdot D_{\mathrm{TV}}(\mu_{\tilde{f}} \| \mu_f)[s] \leq \frac{2\gamma}{1-\gamma} \mathcal{R}_\pi. \tag{73}$$

As a result,

$$\left| \mathop{\mathbb{E}}_{\substack{f \sim p_{\mathrm{train}}(\cdot) \\ \tilde{f} \sim p_{\mathrm{eval}}(\cdot) \\ s \sim d^{\mu_{\tilde{f}}}(\cdot)}} [D_{\mathrm{TV}}(\tilde{\mu}_f \| \mu_f)[s]] - \mathfrak{D}_{\mathrm{train}} \right|$$

$$\leq \int_{\mathcal{F}_{\mathrm{train}}} p_{\mathrm{train}}(f) \int_{\mathcal{F}_{\mathrm{eval}}} p_{\mathrm{eval}}(\tilde{f}) \cdot \left\{ \left| \mathop{\mathbb{E}}_{s \sim d^{\mu_{\tilde{f}}}(\cdot)} [D_{\mathrm{TV}}(\tilde{\mu}_f \| \mu_f)[s]] - \mathop{\mathbb{E}}_{s \sim d^{\mu_f}(\cdot)} [D_{\mathrm{TV}}(\tilde{\mu}_f \| \mu_f)[s]] \right| \right\} \mathrm{d}\tilde{f} \mathrm{d}f$$

$$\leq \int_{\mathcal{F}_{\mathrm{train}}} p_{\mathrm{train}}(f) \int_{\mathcal{F}_{\mathrm{eval}}} p_{\mathrm{eval}}(\tilde{f}) \cdot \left\{ \frac{2\gamma}{1-\gamma} \mathcal{R}_\pi \cdot \max_s D_{\mathrm{TV}}(\tilde{\mu}_f \| \mu_f)[s] \right\} \mathrm{d}\tilde{f} \mathrm{d}f$$

$$= \int_{\mathcal{F}_{\mathrm{train}}} p_{\mathrm{train}}(f) \cdot \left\{ \frac{2\gamma}{1-\gamma} \mathcal{R}_\pi \cdot \max_s D_{\mathrm{TV}}(\tilde{\mu}_f \| \mu_f)[s] \right\} \cdot \int_{\mathcal{F}_{\mathrm{eval}}} p_{\mathrm{eval}}(\tilde{f}) \mathrm{d}\tilde{f} \mathrm{d}f$$

$$= \int_{\mathcal{F}_{\mathrm{train}}} p_{\mathrm{train}}(f) \cdot \left\{ \frac{2\gamma}{1-\gamma} \mathcal{R}_\pi \cdot \max_s D_{\mathrm{TV}}(\tilde{\mu}_f \| \mu_f)[s] \right\} \mathrm{d}f = \frac{2\gamma}{1-\gamma} \mathcal{R}_\pi \int_{\mathcal{F}_{\mathrm{train}}} p_{\mathrm{train}}(f) \cdot \max_s D_{\mathrm{TV}}(\tilde{\mu}_f \| \mu_f)[s] \mathrm{d}f. \tag{74}$$

We previously defined $\sigma_{\mathrm{train}} = \max_{f \in \mathcal{F}_{\mathrm{train}}} \{\max_s D_{\mathrm{TV}}(\tilde{\mu}_f \| \mu_f)[s]\}$, so that

$$\left| \mathop{\mathbb{E}}_{\substack{f \sim p_{\mathrm{train}}(\cdot) \\ \tilde{f} \sim p_{\mathrm{eval}}(\cdot) \\ s \sim d^{\mu_{\tilde{f}}}(\cdot)}} [D_{\mathrm{TV}}(\tilde{\mu}_f \| \mu_f)[s]] - \mathfrak{D}_{\mathrm{train}} \right| \leq \frac{2\gamma}{1-\gamma} \mathcal{R}_\pi \int_{\mathcal{F}_{\mathrm{train}}} p_{\mathrm{train}}(f) \cdot \max_s D_{\mathrm{TV}}(\tilde{\mu}_f \| \mu_f)[s] \mathrm{d}f$$

$$\leq \frac{2\gamma\sigma_{\mathrm{train}}}{1-\gamma} \mathcal{R}_\pi \int_{\mathcal{F}_{\mathrm{train}}} p_{\mathrm{train}}(f) \mathrm{d}f = \frac{2\gamma\sigma_{\mathrm{train}}}{1-\gamma} \mathcal{R}_\pi, \tag{75}$$

thus, the second term is bounded by

$$\mathop{\mathbb{E}}_{\substack{f \sim p_{\mathrm{train}}(\cdot) \\ \tilde{f} \sim p_{\mathrm{eval}}(\cdot) \\ s \sim d^{\mu_{\tilde{f}}}(\cdot)}} [D_{\mathrm{TV}}(\tilde{\mu}_f \| \mu_f)[s]] \leq \frac{2\gamma\sigma_{\mathrm{train}}}{1-\gamma} \mathcal{R}_\pi + \mathfrak{D}_{\mathrm{train}}. \tag{76}$$

Finally, combining (67), (68) and (76), we have

$$\mathfrak{D}_{\mathrm{eval}} \leq \left( 1 + \frac{2\gamma\sigma_{\mathrm{train}}}{1-\gamma} \right) \mathcal{R}_\pi + \mathcal{R}_{\tilde{\pi}} + \mathfrak{D}_{\mathrm{train}}, \tag{77}$$

concluding the proof. $\qquad\square$

# B. A More Detailed Explanation of Our Hypothesis

In Section 4, we claimed that "*The DML loss encourages them to make consistent decisions on the same observations, meaning that any irrelevant features learned by policy A are likely to result in suboptimal performance for policy B, and vice versa.*" Here, we aim to provide a more detailed explanation to help readers better understand this point.

Let's consider a simple environment where the agent is in a rectangular space and attempts to pick up coins to earn rewards (see Figure 6). The agent's observations are the current pixels.

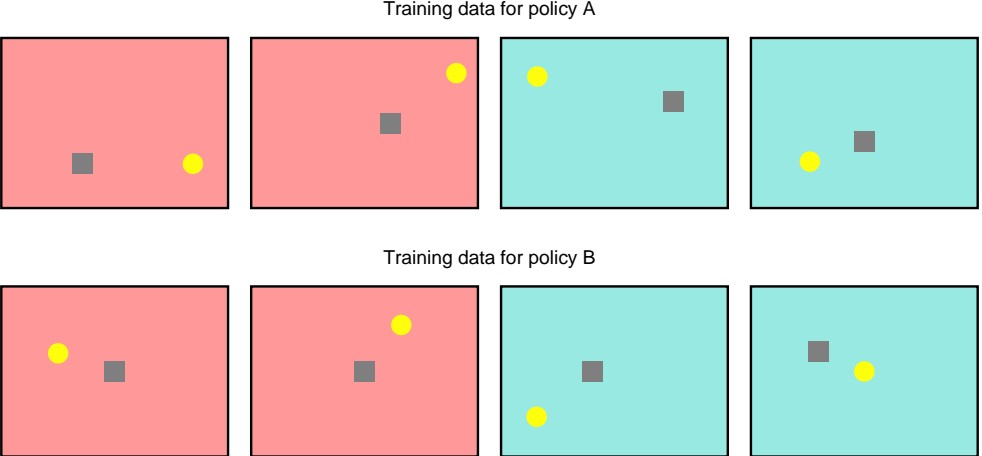

*Figure 6.* This is a simple rectangular environment where the gray agent's goal is to pick up circular coins.

It is clear that the agent's true objective is to pick up the coins, and the background color is a spurious feature. However, upon observing the training data for policy A, we can see that in the red background, the coins are always on the right side of the agent, while in the cyan background, the coins are always on the left side. As a result, when training policy A using reinforcement learning algorithms, it is likely to exhibit overfitting behavior, such as moving to the right in a red background and to the left in a cyan background.

However, the overfitting of policy A to the background color will fail in the training data of policy B, because in policy B's training data, regardless of whether the background color is red or cyan, the coin can appear either on the left or right side of the agent. Therefore, through DML, policy A is regularized by the behavior of policy B during the training process, effectively preventing policy A from overfitting to the background color. In other words, any irrelevant features learned by policy A could lead to suboptimal performance of policy B, and vice versa. Thus, we hypothesize that this process will force both policy A and policy B to learn the true underlying semantics, ultimately converging to meta-representations.

# C. More Empirical Results

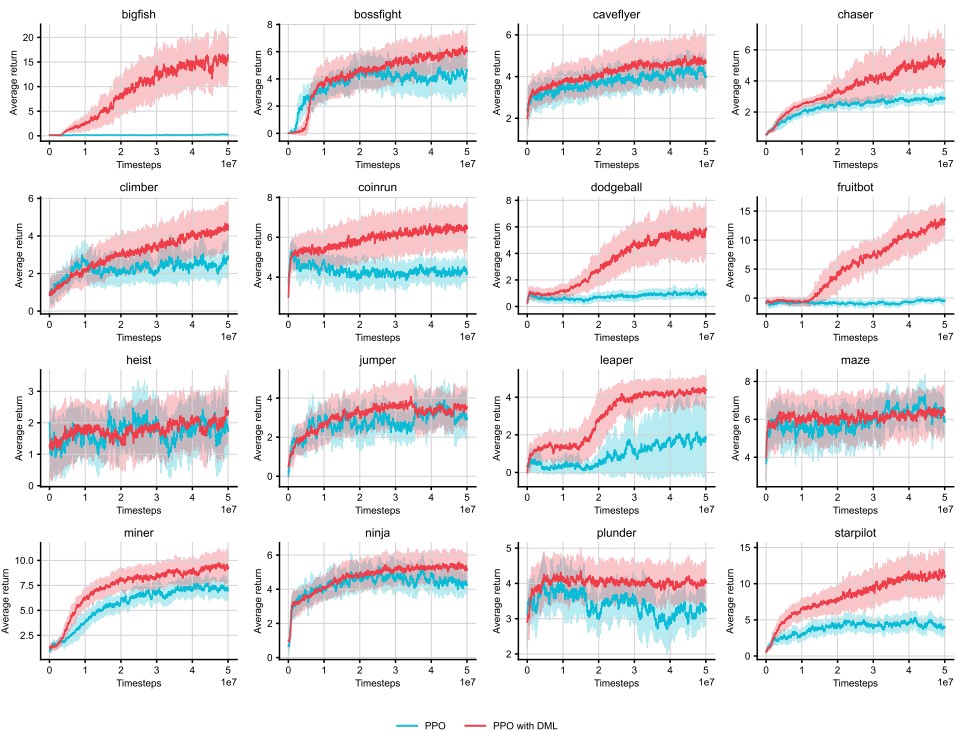

*Figure 7.* Generalization performance from 500 levels in each environment. The mean and standard deviation are shown across 3 seeds.

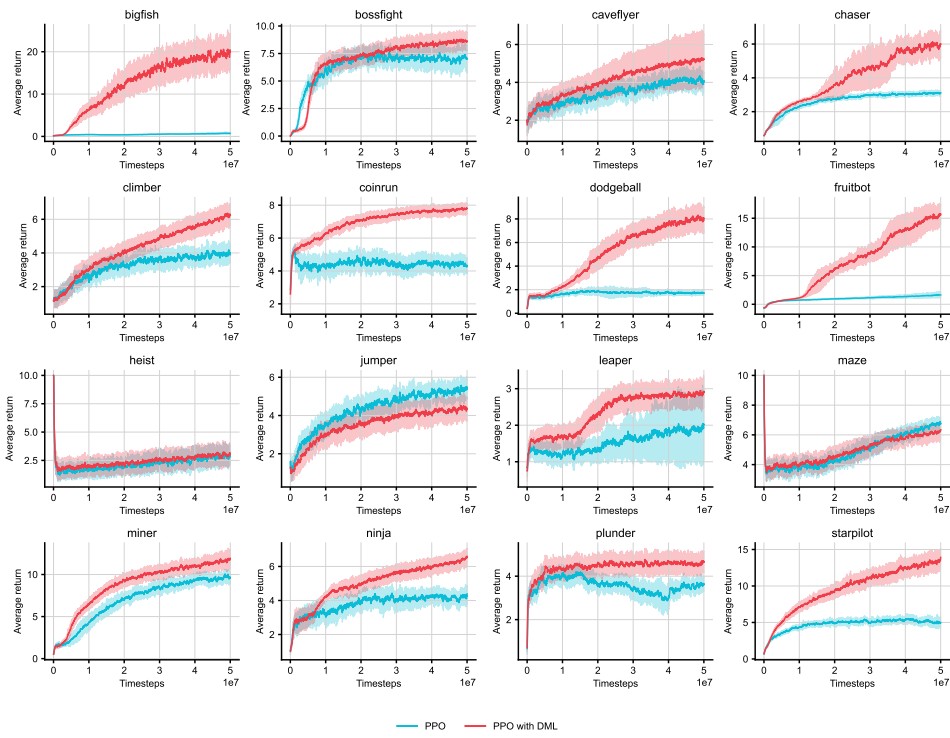

*Figure 8.* Training performance from 500 levels in each environment. The mean and standard deviation are shown across 3 seeds.

---

**Algorithm 3** Proximal Policy Optimization (PPO)

---

1: **Initialize:** Policy and value nets $\pi_\theta$ and $V_\phi$, clipping parameter $\epsilon$, value loss coefficient $c_1$, policy entropy coefficient $c_2$
2: **Output:** Optimal policy network $\pi_{\theta*}$
3: **while** not converged **do**
4:   # Data collection
5:   Collect data $\mathcal{D} = \{(o_t, a_t, r_t)\}_{t=1}^N$ using the current policy network $\pi_\theta$
6:   # The networks before updating
7:   $\pi_{\theta_{\text{old}}} \leftarrow \pi_\theta, \; V_{\phi_{\text{old}}} \leftarrow V_\phi$
8:   # Estimate the advantage $\hat{A}(o_t, a_t)$ based on $V_{\phi_{\text{old}}}$
9:   Use GAE (Schulman et al., 2015) technique to estimate the advantage $\hat{A}(o_t, a_t)$
10:   # Estimate the return $\hat{R}_t$
11:   $\hat{R}_t \leftarrow V_{\phi_{\text{old}}}(o_t) + \hat{A}(o_t, a_t)$
12:   **for** each training epoch **do**
13:     # Compute policy loss $\mathcal{L}_p$
14:     $\mathcal{L}_p \leftarrow -\frac{1}{N} \sum_{t=1}^N \min \left[ \frac{\pi_\theta(a_t|o_t)}{\pi_{\theta_{\text{old}}}(a_t|o_t)} \cdot \hat{A}(o_t, a_t), \text{clip}\left( \frac{\pi_\theta(a_t|o_t)}{\pi_{\theta_{\text{old}}}(a_t|o_t)}, 1 - \epsilon, 1 + \epsilon \right) \cdot \hat{A}(o_t, a_t) \right]$
15:     # Compute policy entropy $\mathcal{L}_e$ and value loss $\mathcal{L}_v$
16:     $\mathcal{L}_e \leftarrow \frac{1}{N} \sum_{t=1}^N \mathcal{H}(\pi_\theta(\cdot|o_t)), \; \mathcal{L}_v \leftarrow \frac{1}{2N} \sum_{t=1}^N [V_\phi(o_t) - \hat{R}_t]^2$
17:     # Compute total loss $\mathcal{L}$
18:     $\mathcal{L} \leftarrow \mathcal{L}_p + c_1 \mathcal{L}_v - c_2 \mathcal{L}_e$
19:     # Update parameters $\theta$ and $\phi$ through backpropagation, $\lambda_\theta$ and $\lambda_\phi$ is the step sizes
20:     $\theta \leftarrow \theta - \lambda_\theta \nabla_\theta \mathcal{L}, \; \phi \leftarrow \phi - \lambda_\phi \nabla_\phi \mathcal{L}$
21:   **end for**
22: **end while**

---

## D. More Implementation Details

### D.1. Proximal Policy Optimization

In our experiments, we employ Proximal Policy Optimization (PPO) as our baseline algorithm. Specifically, given the policy network $\pi_\theta$, the value network $V_\phi$, and any observation-action pair $(o_t, a_t)$, the value loss is

$$\mathcal{L}_v = \frac{1}{2}[V_\phi(o_t) - \hat{R}_t]^2, \tag{78}$$

where $\hat{R}_t$ is the estimated discounted return at step $t$ using the Generalized Advantage Estimation (GAE) (Schulman et al., 2015) technique. And the policy loss is

$$\mathcal{L}_p = -\min \left[ \frac{\pi_\theta(a_t|o_t)}{\pi_{\theta_{\text{old}}}(a_t|o_t)} \cdot \hat{A}(o_t, a_t), \text{clip}\left( \frac{\pi_\theta(a_t|o_t)}{\pi_{\theta_{\text{old}}}(a_t|o_t)}, 1 - \epsilon, 1 + \epsilon \right) \cdot \hat{A}(o_t, a_t) \right], \; \mathcal{L}_e = \mathcal{H}(\pi_\theta(\cdot|o_t)), \tag{79}$$

where $\mathcal{H}(\cdot)$ represents the entropy of the output action distribution. The pseudo-code for PPO is provided in Algorithm 3.

### D.2. PPO with DML

Our approach introduces an additional KL divergence loss to encourage mutual learning between the two agents, which is

$$\mathcal{L}_{\text{DML}} = \mathcal{L}_p + c_1 \mathcal{L}_v - c_2 \mathcal{L}_e + \alpha \mathcal{L}_{\text{KL}}, \tag{80}$$

where $\mathcal{L}_p + c_1 \mathcal{L}_v - c_2 \mathcal{L}_e$ is the reinforcement learning loss, and

$$\mathcal{L}_{\text{KL}} = D_{\text{KL}}(\pi_{\hat{\theta}} \| \pi_\theta) \tag{81}$$

is the KL divergence between the current policy and the other agent's policy, $\alpha$ is the weight, and $\pi_{\hat{\theta}}$ denotes the other agent's policy. Thus, this additional KL loss encourages the two agents to make consistent decisions for the same observations.

## D.3. Hyperparameter Settings

Table 2 shows the detailed hyperparameter settings in our code, with the main hyperparameters consistent with the hard-level settings in Cobbe et al. (2020), except that we trained for 50M steps instead of 200M. We trained the policy on the initial 500 levels and tested its generalization performance across the entire level distribution.

*Table 2.* Detailed hyperparameters in Procgen.

| Hyperparameter\Algorithm | PPO (Schulman et al., 2017) | PPO with DML (ours) |
| --- | --- | --- |
| Number of workers | 64 | 64 |
| Horizon | 256 | 256 |
| Learning rate | 0.0005 | 0.0005 |
| Learning rate decay | No | No |
| Optimizer | Adam | Adam |
| Total interaction steps | 50M | 50M |
| Update epochs | 3 | 3 |
| Mini-batches | 8 | 8 |
| Batch size | 16384 | 16384 |
| Mini-batch size | 2048 | 2048 |
| Discount factor $\gamma$ | 0.999 | 0.999 |
| GAE parameter $\lambda$ | 0.95 | 0.95 |
| Value loss coefficient $c_1$ | 0.5 | 0.5 |
| Entropy loss coefficient $c_2$ | 0.01 | 0.01 |
| Clipping parameter $\epsilon$ | 0.2 | 0.2 |
| KL divergence weight $\alpha$ | - | 1.0 |

