# OpenReview forum: "The Meta-Representation Hypothesis"
_ICML.cc/2025/Conference — Submitted to ICML 2025_

### Official Review · Reviewer_twjt · 2025-02-24

**Overall Recommendation:** 2

**Summary:**

This paper proposes "the meta-representation hypothesis", i.e., learning a representation that reflects the abstract, high-level understanding of inputs can lead to better generalization of RL agents. The paper models the generalization problem in RL using an MDP generator and assumes that all different MDPs the agent interacts with are generated from the same underlying MDP. Theoretically, the paper shows that minimizing the TV distance between the policies in different MDPs could improve the lower bound of the generalization performance. Empirically, the paper shows that deep mutual learning (DML) can regularize the agent's representations, resulting in improved generalization (though I think the main idea is slightly different from the original DML, see "Other comments and suggestions" for details). The proposed method is evaluated on Procgen.

**Post rebuttal update:**
After considering the author's rebuttal and going through all the reviews, I decided to lower my score by 1 due to two reasons: (1) the authors did not provide a convincing clarification of the difference between the proposed "meta-representation" and invariant representations; (2) the main idea of using DML in RL to facilitate generalization turns out to be not new. I also think addressing (1) may require quite a bit of rewriting, which I would let AC and other reviewers decide whether it warrants another round of review.

**Claims And Evidence:**

All claims made in the paper are supported by theoretical or empirical evidence.

**Essential References Not Discussed:**

I think prior work in two related areas should be discussed:

- **Invariant representation learning:** The authors view "meta-representations" as representations that induce the same policy for all MDPs with the same underlying MDP. This formulation is similar to learning invariant representations across multiple training domains, which was first proposed in the supervised learning context (e.g., [1]) and has also been extended to RL [2].

- **Representation learning in RL:** Meta-representations also relate to state representations that discard task-irrelevant information in the agent's raw observations, e.g., via bisimulation metrics [3]. Discussing the work in this area is also necessary.

[1] Arjovsky et al. Invariant risk minimization. 2019.

[2] Sonar et al. Invariant policy optimization: Towards stronger generalization in reinforcement learning. L4DC, 2021.

[3] Learning invariant representations for reinforcement learning without reconstruction. ICLR, 2021.

**Experimental Designs Or Analyses:**

The main experiments are conducted on Procgen, which is a standard benchmark for RL generalization. The authors mainly compare their method with a standard PPO baseline, which is acceptable in my opinion since the authors do not claim to achieve state-of-the-art performance but mostly focus on demonstrating the effectiveness of DML. Visualizations and ablations are sufficient.

**Methods And Evaluation Criteria:**

The proposed method and evaluation criteria all make sense to me.

**Other Comments Or Suggestions:**

In the original DML paper, different models are trained on the data with the _same_ distribution. Yet in this work, different agents mostly sample different MDPs with _different_ state distributions (this is also reflected in the motivating example in Appendix B). The authors may consider discussing this point in more detail.

**Other Strengths And Weaknesses:**

**Strengths:** The paper is well-written.

**Weaknesses:** The theoretical part is a little hard to follow. In particular, I do not really get the necessity of introducing the first-order approximation $L_{\pi}$ as in TRPO.

**Questions For Authors:**

- What is the relation between "meta-representations" and invariant representations in the literature?

**Relation To Broader Scientific Literature:**

In my opinion, the claimed main contributions of the paper are twofold:

- Theoretical analysis showing that robustness to irrelevant features can facilitate RL generalization.
- Empirical analysis showing that DML improves RL generalization.

Given the naturality of the first point, I think its contribution to the broader scientific literature is a bit limited---quite similar ideas have been proposed and at least partially analyzed in related areas such as invariant representation learning and representation learning in RL (see "Essential references not discussed" for more details).

I think the second point is more interesting---although DML itself is not new, to my knowledge, applying similar ideas to RL by simultaneously training multiple agents and letting them mutually regularize each other's policies to improve generalization is novel.

**Theoretical Claims:**

I briefly skimmed through the proofs and did not find any major problems.

---

> ### Author Rebuttal · Authors · 2025-03-31
>
> Dear Reviewer twjt,
>
> Thank you for your positive assessment of our work. Below, we will address your concerns.
>
> >I think prior work in two related areas should be discussed:
> >- **Invariant representation learning** [1, 2].
> >- **Representation learning in RL** [3].
>
> Thank you for your valuable suggestion. As the rebuttal process does not permit submitting revised PDF files, we will address these two aspects in the related work section of our future extended version.
>
> >The theoretical part is a little hard to follow. In particular, I do not really get the necessity of introducing the first-order approximation $L_ {\pi}$ as in TRPO.
>
> We would be happy to briefly explain for you. We first define the training performance
>
> $$\eta(\pi)=\frac{1}{1-\gamma}\mathbb{E}_ {f\sim p_{\mathrm{train}}(\cdot),s\sim d^{\mu_f}(\cdot),a\sim\mu_f(\cdot|s)}[r(s,a)].$$
>
> Next, according to the performance difference lemma, we have
>
> $$\eta(\tilde{\pi})=\eta(\pi)+\frac{1}{1-\gamma}\mathbb{E}_ {f\sim p_{\mathrm{train}}(\cdot),s\sim d^{\tilde{\mu}_f}(\cdot),a\sim\tilde{\mu}_f(\cdot|s)}[A^{\mu_f}(s,a)].$$
>
> Therefore, our objective is to obtain an updated policy $\tilde{\pi}$ based on $\pi$. However, the state distribution $s\sim d^{\tilde{\mu}_f}(\cdot)$ and action distribution $a\sim\tilde{\mu}_f(\cdot|s)$ in their performance difference are all sampled from the new policy $\tilde{\pi}$, where $\tilde{\mu}_f(\cdot|s)=\tilde{\pi}(\cdot|f(s)),\forall f\in\mathcal{F}$, which is clearly infeasible since $\tilde{\pi}$ is unknown at this stage.
>
> Therefore, we must approximate the state distribution (replace $\tilde{\pi}$), which naturally leads to
>
> $$L_ {\pi}(\tilde{\pi})=\eta(\pi)+\frac{1}{1-\gamma}\mathbb{E}_ {f\sim p_{\mathrm{train}}(\cdot),s\sim d^{\mu_f}(\cdot),a\sim\tilde{\mu}_f(\cdot|s)}[A^{\mu_f}(s,a)].$$
>
> >I think the second point is more interesting---although DML itself is not new, to my knowledge, applying similar ideas to RL by simultaneously training multiple agents and letting them mutually regularize each other's policies to improve generalization is novel.
>
> >In the original DML paper, different models are trained on the data with the _same_ distribution. Yet in this work, different agents mostly sample different MDPs with _different_ state distributions (this is also reflected in the motivating example in Appendix B). The authors may consider discussing this point in more detail.
>
> Thank you for your insightful comments and suggestions—this is indeed the core motivation and a key distinction from the original DML. We will highlight this distinction in subsequent versions by further expanding Appendix B.
>
> >What is the relation between "meta-representations" and invariant representations in the literature?
>
> This is a good question. As you mentioned in the two papers [2, 3], similar concepts have indeed been explored. However, while these approaches typically decouple the generalization problem into robust representation learning (encoder $\phi$) and downstream policy learning, our method is **theoretically and empirically end-to-end.**
>
> Note that our entire paper never introduces an upstream encoder $\phi$, and we find our approach both more elegant and simpler, as it removes the need to decouple the encoder $\phi$'s learning process from the reinforcement learning process, everything is end-to-end.
>
> Ultimately, while meta-representations and invariant representations are conceptually similar, we argue that **the notion of meta-representations is more general**. This is because the agent does not need to explicitly learn a robust encoder; instead, the entire end-to-end policy only needs to improve robustness to irrelevant features. Consequently, components beyond the upstream encoder (such as downstream MLPs in the policy network's architecture) may also contribute to this robustness.
>
> Best,
>
> Authors
>
> ---
> *Reference:*
>
> [1] M Arjovsky et al. Invariant risk minimization.
>
> [2] A Sonar et al. Invariant policy optimization: Towards stronger generalization in reinforcement learning.
>
> [3] A Zhang et al. Learning invariant representations for reinforcement learning without reconstruction.

---

> > ### Comment · Reviewer_twjt · 2025-04-03
> >
> > Thank you for your response. I found the clarification on the theory part helpful.
> >
> > However, honestly, I am not satisfied with the response regarding the comparison between "meta-representations" and invariant representation learning.
> >   - As you mentioned, end-to-end training is appealing, yet it does not deny that these two terms are conceptually similar under your current definition---whether using an explicit encoder or not is only at an implementation level.
> >   - This point appears to be quite important given that you have heavily emphasized the conceptual/philosophical value of "meta-representations" all over the paper.
> >   - In fact, even if you add more discussion with the invariant representation learning methods to the related work section, I still feel that the overall presentation can be quite misleading to readers who are not familiar with the invariant representation learning literature, which I am not sure if it can be addressed without quite a bit rephrasing.
> >
> > I will also raise this point in the reviewer-AC discussion phase.
> >
> > Also, I noticed that you quoted my review in your response to Reviewer ByPb. It turns out that I indeed missed [1] as one of the closely related works. To avoid further misunderstanding, I will also leave a brief clarification there. Sorry for the inconvenience.
> >
> > ---
> >
> > [1] Zhao, Chenyang, and Timothy Hospedales. Robust domain randomised reinforcement learning through peer-to-peer distillation. ACML, 2021.

---

> > > ### Author Response · Authors · 2025-04-03
> > >
> > > Dear Reviewer twjt,
> > >
> > > Thank you for your additional comments. Below, we will provide a comprehensive response to your feedback.
> > >
> > > >As you mentioned, end-to-end training is appealing, yet it does not deny that these two terms are conceptually similar under your current definition---whether using an explicit encoder or not **is only at an implementation level**.
> > >
> > > Thank you for your response. However, the concept of meta-representation differs from invariant representation not only at the implementation level. To clarify this point, we refer to several papers on invariant representation learning [2, 3] and formally define the concept of invariant representation as follows:
> > >
> > > In the framework of invariant representation learning, we define the policy as $\Delta_ {\mathcal{A}}=\pi(o_ t)$, where $o_ t$ is observation and $\Delta_ {\mathcal{A}}$ is the probability distribution over the action space $\mathcal{A}$, is obtained as a composition $\pi=h\circ g$.
> > >
> > > We can regard $z=g(o_t)$ as a learned representation of $o_ t$, and a smaller mlp $\Delta_ {\mathcal{A}}=h(z)$, predicting probability distribution $\Delta_ {\mathcal{A}}$ given representation $z$, both of which are shared across domains.
> > >
> > > Using this framework, we can strive to learn an "invariant" representation $z$ across the source domains, with the hope of achieving better generalization to the target domain. Therefore, invariant representation learning emphasizes the robustness of the upstream encoder $g$ across different domains, i.e.,
> > >
> > > $\min_ {g}\Vert g(o)-g(\tilde{o})\Vert,$
> > >
> > > where $o$ and $\tilde{o}$ represent observations of the same underlying state from different domains.
> > >
> > > For meta-representations, formally, we do not decouple the policy $\pi$ to $h\circ g$. Instead, we directly emphasize the robustness of $\pi$ to $o$ and $\tilde{o}$, i.e.,
> > >
> > > $\min_ {\pi}\Vert\pi(\cdot|o)-\pi(\cdot|\tilde{o})\Vert.$
> > >
> > > In summary, meta-representation focuses on the invariance of the output (see the explanation of Definition 3.6 in our paper), while invariant representation learning mainly emphasizes learning an invariant representation $z$.
> > >
> > > We acknowledge that meta-representations share conceptual similarities with invariant representations. However, to highlight the subtle distinctions between them (as discussed above), we introduced the term "meta-representation." We apologize for any confusion this may have caused.
> > >
> > > >...I still feel that the overall presentation can be quite misleading to readers who are not familiar with the invariant representation learning literature...
> > >
> > > Thank you for your valuable comments! To avoid any misleading implications for readers who are not familiar with invariant representation learning, we will definitely **highlight the subtle distinction** between meta-representation and invariant representation in the related work section.
> > >
> > > >It turns out that I indeed missed [1] as one of the closely related works...
> > >
> > > We sincerely appreciate your thorough review. Indeed, [1] is relevant to our work, and we will add it to the related work section in the revised version. We did read [1] and recognize some methodological similarities with our approach, yet we note **four** key differences:
> > >
> > > - **Theoretical Contribution:** [1] primarily provides _empirical_ evidence for the effectiveness of mutual distillation, without theoretical analysis of _why_ it works. In contrast, we prove that generalization performance benefits from policy robustness to irrelevant features, which is a strong theoretical contribution.
> > > - **A Stronger Insight:** In [1], the authors argue that mutual distillation facilitates knowledge sharing between students. However, we demonstrate that DML actually enhances the student's robustness to irrelevant features. We provide strong evidence for this statement using t-SNE and random CNNs.
> > > - **Domain Sampling Mechanism:** In [1], the authors argue that training across multiple domains may lead to high variance, so in each iteration, each student $\pi_ i$ collects data only from a specific domain $\xi_ i$. However, we demonstrate that students can indeed be trained across multiple domains, with DML serving as a form of mutual regularization.
> > > - **Harder Generalization:** In [1], the domain distribution is defined by the authors, whereas in our experiments, it is unknown, making generalization even more challenging.
> > >
> > > In summary, we acknowledge the relevance of these two works, but we also emphasize the additional contributions our work makes to the generalization of reinforcement learning, particularly the non-trivial theoretical analysis.
> > >
> > > Best,
> > >
> > > Authors
> > >
> > > ---
> > > *Reference:*
> > >
> > > [1] C Zhao et al. Robust domain randomised reinforcement learning through peer-to-peer distillation.
> > >
> > > [2] AT Nguyen et al. Domain invariant representation learning with domain density transformations.
> > >
> > > [3] H Qi et al. Data-driven offline decision-making via invariant representation learning.

---

### Official Review · Reviewer_Rz5E · 2025-03-13

**Overall Recommendation:** 3

**Summary:**

The paper deals with RL environments where the observations presented to an agent are noisy transformations of the true state via a rendering function that is drawn from an environment dependent distribution.

This paper first rigorously demonstrates how generalization performance of RL agents can suffer if the features they learn to use are not robust to noise. It is theoretically shown that the lower bound of the generalization performance can be improved by improving robustness.

The paper then proposes that Deep Mutual Learning can help introduce this robustness. This is based on intuition, and is then demonstrated empirically in various environments.

## Update after review
Rating unchanged. I believe this is a borderline accept. I believe the evaluation is not comprehensive enough for a higher rating.

**Claims And Evidence:**

Claim 1:
The generalization performance of a policy under the specific setting of the randomly sampled rendering function (required to be a bijection) is affected by the robustness of the policy to changes in the rendering function.

This is proven theoretically.

Claim 2:
Deep Mutual Learning induces more robust features that help generalization.

This is supported by a variety of experiments that show how the policy's performance does not degrade as rapidly under noisy observations as it would without DML. The experiments cover a variety of environments.

The only concern I have here is that the evaluation is limited to only specific kinds of distributions of rendering functions.

**Essential References Not Discussed:**

N/A

**Experimental Designs Or Analyses:**

The experiments and analyses are reasonably designed and are a good way to test the validity of the claims. However, I found the choice of rendering functions tested in the experiments to be very limited. See section on strengths and weaknesses for details.

**Methods And Evaluation Criteria:**

Methods:
The proposed method is not novel - it is an existing method.

Evaluation:
The evaluation is sound and covers two key aspects:
1. Robustness - The primary metric is the divergence of the policy when the observations that the agent gets for the same state are changed by addition of noise.
2. Performance - The evaluation criteria also includes the average performance of the learned policies under different rendering functions, specifically under different addition of noise through randomized Gaussian filters.

**Other Comments Or Suggestions:**

N/A

**Other Strengths And Weaknesses:**

The paper does rigorously show how robustness to changes in the rendering function can affect performance, and why it is important to be tolerant to such changes. This is a useful result.  The experiments also nicely show how DML can help tackle some of these changes. This evaluation is a good benchmark to check if an RL method is robust to some noise.

I, however, found that the evaluation in the paper is limited to a narrow set of rendering function changes. I believe this limits the significance of the experiments in the paper. Major changes in the rendering function are not considered. E.g. what if in a game the sprite is changed across episodes? In this sense, I find that the experiments are talking more about robustness to noise, rather than generalization across a wide variety of rendering functions. The paper does allude to generalization of this kind in all sections except the experiment section.

**Questions For Authors:**

1. Can this be evaluated under a broader distribution of rendering functions?

**Relation To Broader Scientific Literature:**

This paper is loosely related to the broader literature on meta-representations. The paper is more closely related to works on robust policy learning. The paper presents a proof of a widely held intuition that robustness to noisy observations improves performance. This is novel in my knowledge. There are no novel methods proposed in the paper based on the claim. However, an existing method (Deep Mutual Learning) is evaluated on the criteria shown in the paper.

**Theoretical Claims:**

Did not find any issues in the theoretical results and their proofs.

---

> ### Author Rebuttal · Authors · 2025-03-31
>
> Dear Reviewer Rz5E,
>
> Thank you for your positive feedback on our paper. Below, we will address your concerns.
>
> >The only concern I have here is that the evaluation is limited to only specific kinds of distributions of rendering functions.
>
> >I, however, found that the evaluation in the paper is limited to a narrow set of rendering function changes. I believe this limits the significance of the experiments in the paper. Major changes in the rendering function are not considered. E.g. what if in a game the sprite is changed across episodes? In this sense, I find that the experiments are talking more about robustness to noise, rather than generalization across a wide variety of rendering functions. The paper does allude to generalization of this kind in all sections except the experiment section.
>
> >Can this be evaluated under a broader distribution of rendering functions?
>
> Thank you for your insightful comments. However, the observation that our experiments were evaluated under limited rendering functions might be a misunderstanding.
>
> Please note that the Procgen environment samples from a nearly infinite pool of levels when testing generalization performance (each level can be considered as a specific rendering function, since the task semantics remain unchanged). We would like to quote the first paragraph of Section 2 in the original paper of Procgen [1]:
>
> >_The Procgen Benchmark consists of 16 unique environments designed to measure both sample efficiency and generalization in reinforcement learning. These environments greatly benefit from the use of procedural content generation—the algorithmic creation of **a near-infinite supply** of highly randomized content. In these environments, employing procedural generation is far more effective than relying on fixed, human-designed content._
>
> You may also refer to our anonymous website at https://dml-rl.github.io/. In the Coinrun environment specifically, the sprite do change across different episodes. Our generalization curves are indeed tested through sampling from a **near-infinite set of rendering functions:** during training, agents only have access to a limited set of rendering functions, i.e., the first 500 levels, while generalization performance is evaluated across an **infinite number of levels**. We hope this addresses your core concerns.
>
> Best,
>
> Authors
>
> ---
> *Reference:*
>
> [1] K Cobbe et al. Leveraging procedural generation to benchmark reinforcement learning.

---

### Official Review · Reviewer_ByPb · 2025-03-17

**Overall Recommendation:** 2

**Summary:**

The paper proposes to combine Deep Mutual Learning with RL. In Deep Mutual Learning, several learners learn independently but at the same try to minimize the KL between their predictive distributions. The paper hypothesizes that two RL policies can learn from different MDPs — where each MDP has its own randomly sampled observation function while the policies try to minimize the KL between them. This would lead to the learning of robust representation functions. The randomly perturbed observation function is a key aspect of the paper — in their paper they apply a CNN with random weights to the observation to map the true observation to a perturbed one. The paper tests this hypothesis via PPO and shows that Deep Mutual Learning is helpful for generalization on the Procgen Benchmark.

**Claims And Evidence:**

Yes.

**Essential References Not Discussed:**

[1] Zhao, Chenyang, and Timothy Hospedales. "Robust domain randomised reinforcement learning through peer-to-peer distillation."

**Experimental Designs Or Analyses:**

Yes, the experimental design is sound.

**Methods And Evaluation Criteria:**

Yes.

**Other Comments Or Suggestions:**

Resolve the Cons.

**Other Strengths And Weaknesses:**

## Pros

1. Tackles an important problem about having a robust perception function for RL.
2. A positive thing is that the whole model is learned end-to-end via RL rather than separately learning a representation model.

## Cons

1. The idea of RL + DML appears to be not very novel as a similar thing is explored here [1]. Therefore I believe the key novelty is mainly about how one can get perturbed MDPs. However, I tend to think that the randomized CNN approach is a bit too simplistic. See my comment in the “Questions for Authors” section.
2. Only evaluates PPO. It could help to have a few other RL algorithms.
3. Why are existing representation learning approaches not worthy candidates for comparison as encoders? Just seeking clarification rather than asking for additional baselines. For instance, there are many representation learning methods that can be considered to learn robust features e.g., approaches like SimCLR [2, 3] use data augmentation by having jitter, distortions, etc.
4. It would be useful to analyze the informativeness of the meta-representation. It is possible that the representation is a lossy one. This isn’t necessarily a bad thing, just that some insight about it needs to be shared with the community — and how lossy this representation e.g., does it simply erase the textural information from the representation while retaining only edge features? If yes, then tasks that rely on textural information information might suffer.
5. More ablation could help. A simple ablation is to mix the MDPs during training without using mutual learning. I would be curious what effect that has or what difference that would have with the proposed one.

[1] Zhao, Chenyang, and Timothy Hospedales. "Robust domain randomised reinforcement learning through peer-to-peer distillation."

[2] Chen, Ting, et al. "A simple framework for contrastive learning of visual representations.”

[2] Agarwal, Rishabh, et al. "Contrastive behavioral similarity embeddings for generalization in reinforcement learning.”

**Questions For Authors:**

Are there ideas about how to go beyond just CNN-weight randomization for sampling the $f$s? The kind of variations we see in the real world are more systematic in nature: changes in lighting, camera angle, and shininess. Or it could be factor changes e.g., size or color changes. Many of these cannot be achieved by CNN-weight randomization. On a related note, are there any thoughts on how this kind of MDP variation can be obtained in real-world data?

**Relation To Broader Scientific Literature:**

See my comments in the "Cons" section below.

**Theoretical Claims:**

Yes, there are theoretical claims, but I have not checked them thoroughly due to my broader concerns about novelty.

---

> ### Author Rebuttal · Authors · 2025-03-31
>
> Dear Reviewer ByPb,
>
> Thank you for your careful evaluation on our paper. Below, we will address your concerns.
>
> >The idea of RL + DML appears to be not very novel. Therefore I believe the key novelty is mainly about how one can get perturbed MDPs. However, I tend to think that the randomized CNN approach is a bit too simplistic.
>
> The concept of DML is not novel and was used in supervised learning. However, although our core algorithm is based on existing methods, the DML used in this paper is **with strong motivation** and different from the original one. We would like to quote reviewer twjt's comment: "_Although DML itself is not new, to my knowledge, applying similar ideas to RL by simultaneously training multiple agents and letting them mutually regularize each other's policies to improve generalization is novel._"
>
> Moreover, we conducted an in-depth analysis of why DML is effective and provided both theoretical and empirical evidence demonstrating its generalization benefits, thereby constituting our additional contribution.
>
> >Only evaluates PPO. It could help to have a few other RL algorithms.
>
> Thank you for your suggestion. We have added two additional baselines, SPO and PPG. We have the following generalization performance results:
>
> | Algorithm   | bigfish | dodgeball  | fruitbot | starpilot |
> |--------|-----|------|------|------|
> | SPO   | $1.16\pm1.03$| $1.74\pm1.0$| $2.21\pm1.82$ | $6.16\pm1.42$ |
> | SPO with DML   | $5.44\pm2.92$| $5.22\pm1.57$ |$1.12\pm1.02$ | $8.43\pm2.19$ |
>
> Considering that PPG is an enhanced version of PPO that incorporates additional distillation strategies with relatively complex implementation, we specifically compare the generalization performance between standard PPG and PPG utilizing the frozen encoder obtained from PPO with DML:
>
> | Algorithm   | bigfish | dodgeball  | fruitbot | starpilot |
> |--------|-----|------|------|------|
> | PPG (original)    | $11.67\pm5.71$| $6.43\pm1.65$| $20.23\pm1.79$ | $11.97\pm3.2$ |
> | PPG (DML encoder)   | $30.0\pm4.95$| $9.28\pm2.5$ |$19.07\pm1.82$ | $18.97\pm4.27$ |
>
> **More details:** https://anonymous.4open.science/r/meta-hypothesis-rebuttal-C70B/README.md
>
> >Why are existing representation learning approaches not worthy candidates for comparison as encoders?
>
> Thanks. Indeed, there are numerous works based on data augmentation. However, data augmentation introduces additional prior knowledge, which inevitably induces biases in the training data. These biases may be either beneficial or harmful to generalization, and they require human prior knowledge to guide the selection of appropriate data augmentation techniques for the given scenario.
>
> Therefore, our approach emphasizes spontaneous learning of the underlying semantics by agents through mutual regularization, which is simple to implement, intrinsically unbiased, and generally applicable to a wide range of scenarios. More importantly, while prior works decouple the generalization problem into robust representation learning and downstream policy learning, our method is **theoretically and empirically end-to-end.**
>
> >It would be useful to analyze the informativeness of the meta-representation... Does it simply erase the textural information from the representation while retaining only edge features? ...
>
> Thank you for your constructive insights. This depends on specific scenarios. The total loss consists of both RL loss and KL regularization loss. If texture features are important for the current task, the agent will learn to preserve them, as removing such features would make the RL loss term harder to optimize, potentially resulting in suboptimal policy performance.
>
> >More ablation could help. A simple ablation is to mix the MDPs during training without using mutual learning...
>
> Thank you for your suggestion. However, directly mixing the training data from both agents may not be theoretically sound, as PPO's reinforcement learning loss requires data sampled from the old policy for computation, and the data distributions sampled by the two policies could differ. Therefore, we **have included an ablation study** with a PPO baseline that uses double the batch size and number of interactions. Due to response length limitations, please refer to the table in our response to reviewer bqXb and our link.
>
> >Are there ideas about how to go beyond just CNN-weight randomization for sampling the $f$s?
>
> We appreciate the insightful feedback. Indeed, the real-world variations are challenging to capture via CNN randomization alone. We also **tested the robustness of DML under different brightness, contrast, and hue conditions**, see https://anonymous.4open.science/r/meta-hypothesis-rebuttal-C70B/README.md
>
> For real-world MDP variation, we could consider: (a) Vary factors (e.g., lighting/angle) in real-world datasets using robotic platforms, e.g., habitat. (b) Leverage physics-aware tools, e.g., Omniverse, to simulate plausible variations (e.g., shadows) on real data.
>
> Best,
>
> Authors

---

### Official Review · Reviewer_bqXb · 2025-03-23

**Overall Recommendation:** 4

**Summary:**

The paper tackles generalisation in reinfocement learning introducing the concept of "meta-representation", which is an abstract representation of a state shared by all instances with shared semantics, and separated from the details of a particular high-dimensional observation.

The paper brings two (almost separated) contributions:
1. A theoretical claim around a theory of generalization in reinforcement learning. By proving a couple of bounds, the authors make a formal argument for their claim that policy robustness to irrelevant features contributes to improved generalization.

2. A hypothesis that deep mutual learning helps learning more robust features, improving generalisation across environments sharing the same semantics. A set of experiments show better generalisation when DML is applied on top of a popular Deep RL algorithm (PPO).

**Claims And Evidence:**

1. The central hypothesis about DML  helping learning meta-representations, threfore improving generalization performance is supported by experiments on top of a PPO baseline in the Procgen environment.

Two additional experiments further support the hypothesis:
1. random convolutional features to demonstrate the variance of a trained encoder to noisy observations
2. retraining policies on top of frozen encoders

**Essential References Not Discussed:**

Not aware of anything major missing.

**Experimental Designs Or Analyses:**

Did you correct for the "actual" number of environment interactions?
Does the DML combo with two policies see double the states compared to the PPO baseline?

An ablation for the number of parameters would also make a more compelling argument.

**Methods And Evaluation Criteria:**

ProcGen is an appropriate benchmark to measure generalisation in RL.
PPO represents a good baseline.

**Other Comments Or Suggestions:**

I won't comment on the philosophical bits in the paper, but a process is not an insight. Maybe rephrase that.

**Other Strengths And Weaknesses:**

Strengths
1. Careful formalisation of the problem and the central claims.
2. Experiments clearly support the hypothesis in the context of PPO.

Weaknesses
1. Limited evaluation (a single algorithm as baseline, a single set of environments).

**Questions For Authors:**

1. The theory assumes a bijection between the real state space and the observation space, which excludes many cases where partial observability collapses multiple states under the same observation. Could you comment on how your theory would extend to Partial observability, or how important the assumption of such a bijection is beyond formalising the problem?

2. Could you add an ablation for the number of parameters? PPO+DML trains two models (double the parameters) compared to the baseline, doesn't it?
3. Could you add an ablation for the number of steps (or just reuse the data to make a plot with the number of transitions performed on the x axis)?

**Relation To Broader Scientific Literature:**

Section 6 lists the relevant works, although discussing in more detail what each reference brings would be a better way to present them than just enumerating a long list.
Although it's understandable given the paper length constraints, section 6 could be expanded and would make the submission stronger.

**Theoretical Claims:**

I did check the correctness of the proofs (also reviewed the supplementary material) and found no errors.

---

> ### Author Rebuttal · Authors · 2025-03-31
>
> Dear Reviewer bqXb,
>
> Thank you for your constructive feedback on our paper! Below, we will address your concerns.
>
> >Did you correct for the "actual" number of environment interactions? Does the DML combo with two policies see double the states compared to the PPO baseline?
>
> Thanks. First, the two policies indeed double the actual number of interactions. However, when computing their respective losses, the two agents **cannot** directly access each other's training data for RL training. Specifically, assume that agent A (denote as $\pi_ A$) collects a batch of training data $\mathcal{D}_ A=\textbraceleft(o_ 1^A,a_ 1^A,r_ 1^A),\dots,(o_ k^A,a_ k^A,r_ k^A)\textbraceright$, agent B (denote as $\pi_ B$) collects a same batch of training data $\mathcal{D}_ B=\textbraceleft(o_ 1^B,a_ 1^B,r_ 1^B),\dots,(o_ k^B,a_ k^B,r_ k^B)\textbraceright$. Then, the total loss for agent A is
>
> $\mathcal{L}_ {A}=\underbrace{\mathcal{L}_ {\mathrm{RL}}(\mathcal{D}_ A)}_ {\mathrm{\text{RL loss}}}+\underbrace{\frac{1}{k}\sum_ {i=1}^{k}D_ {\mathrm{KL}}(\pi_ B(\cdot|o_ i^A)\Vert\pi_ A(\cdot|o_ i^A))}_ {\mathrm{\text{DML loss}}}$
>
> Here, agent A's total loss $\mathcal{L}_ {A}$ only involves agent A's **own** dataset $\mathcal{D}_ A$, while the DML loss serves as a regularization term. Therefore, from each agent's own perspective, the number of interactions remains consistent with the baseline.
>
> >Could you add an ablation for the number of parameters?
>
> >Could you add an ablation for the number of steps...?
>
> **We have added the ablation experiments with double parameters.** To ensure fairness, we also doubled the batch size and total number of interactions for the PPO baseline. The table below shows the generalization performance across four environments:
>
> | Algorithm   | bigfish | dodgeball  | fruitbot | starpilot |
> |--------|-----|------|------|------|
> | PPO (original)   | $0.26\pm0.23$| $0.92\pm0.46$| $-0.5\pm0.81$ | $3.99\pm1.21$ |
> | PPO (double parameters, batch size and interactions)   | $3.99\pm4.06$| $1.94\pm0.81$ | $8.58\pm3.99$ | $3.18\pm1.33$ |
> | PPO with DML (ours)   | $16.11\pm4.63$| $5.66\pm1.98$ | $13.23\pm3.04$ | $11.28\pm3.04$ |
>
> >Limited evaluation.
>
> **We provided an additional RL baseline** with our method, please see Fig.2 in our link.
>
> **For more results**: https://anonymous.4open.science/r/meta-hypothesis-rebuttal-C70B/README.md
>
> >Section 6 lists the relevant works, although...
>
> Thanks! We will definitely expand the related work in Section 6 in our extended paper.
>
> >The theory assumes a bijection between the real state space and the observation space, which excludes many cases where partial observability collapses multiple states under the same observation. Could you comment on how your theory would extend to Partial observability, or how important the assumption of such a bijection is beyond formalising the problem?
>
> Great question! Mathematically, the POMDP problem is consistent with the theoretical framework presented in this paper. Under the POMDP setting, $s$ represents the global information, while $f$ can be regarded as a masking function that obscures the global state $s$ into partial observations $o=f(s)$. Therefore, if we regard the *underlying state* $s$ in this work as *global information* and the *rendering function* $f$ as a *masking function*, then the generalization problem can be viewed as a POMDP problem.
>
> However, for the POMDP problem, $f$ can only be guaranteed to be surjective but not bijective. For example, for two different global information states $s_ 1$ and $s_ 2$, after applying a certain masking function $f$, $f(s_ 1)=f(s_ 2)$ could happen. In this case, $|\mathcal{O}_ f|<|S|$. Just like two similar maze environments, the global states $s_1\neq s_2$. However, by masking the dissimilar parts of the mazes, the agent's observations become identical. This results in the policy being unable to truly distinguish between these two different global states, i.e., $\pi(\cdot|f(s_1))\equiv\pi(\cdot|f(s_2))$, thus potentially failing to learn the optimal policy.
>
> As a result, the PODMP problem is more challenging than the generalization problem in this paper, as the policy may require additional structures to facilitate learning, such as using a recurrent neural network (RNN) to encode historical information, which inherently introduces **non-Markov property**.
>
> Best,
>
> Authors

---

### Decision · Program_Chairs · 2025-05-01

**Decision:**

Reject

**Comment:**

The paper investigates an interesting research direction. However, several concerns remain:
- Reviewer ByPb highlights that this contribution might not be very novel given for instance the following paper that was not cited: [1] Zhao, Chenyang, and Timothy Hospedales. "Robust domain randomised reinforcement learning through peer-to-peer distillation.". This opinion is also confirmed by Reviewer twjt0 in a post-rebuttal comment.
- Reviewer ByPb highlights that limited baselines are considered: "Why are existing representation learning approaches not worthy candidates for comparison as encoders?".
- Reviewer twjt highlights that there is generally a lack of prior work discussion in domains of "Invariant representation learning" and "Representation learning in RL".